# Arctic clouds in ECHAM6 and their sensitivity to cloud microphysics and surface fluxes

Jan Kretzschmar[1], Marc Salzmann[1], Johannes Mülmenstädt[1], and Johannes Quaas[1]

[1]Institute for Meteorology, Universität Leipzig, Vor dem Hospitaltore 1, 04103 Leipzig, Germany

**Correspondence:** Jan Kretzschmar (jan.kretzschmar@uni-leipzig.de)

**Abstract.** Compared to other climate models, the MPI-ESM/ECHAM6 is one of the few models that is able to realistically simulate the typical two-state radiative structure of the Arctic boundary layer and also is able to sustain liquid water at low temperatures as it is often observed in high latitudes. To identify processes in the model that are responsible for the above mentioned features, we compare cloud properties from ECHAM6 to observations from CALIPSO-GOCCP using the COSP satellite simulator and perform sensitivity runs. The comparison shows that the model is able to reproduce the spatial distribution and cloud amount in the Arctic to some extent, but a positive bias in cloud fraction is found in high latitudes, which is related to an overestimation of low- and high-level clouds. We mainly focus on low-level clouds and show that the overestimated cloud amount is connected to surfaces that are covered with snow or ice and is mainly caused by an overestimation of liquid containing clouds. The overestimated amount of Arctic low-level liquid clouds can be related to insufficient efficiency of the Wegener-Bergeron-Findeisen (WBF) process but revising this process alone is not sufficient to improve cloud phase on global scale as it also introduces a negative bias over oceanic regions in high latitudes. Additionally, this measure transformed the positive bias in low-level liquid clouds into a positive bias of low-level ice clouds, keeping the amount of low-level clouds almost unchanged. To avoid this spurious increase in ice clouds, we allowed for supersaturation with respect to ice using a temperature weighted scheme for saturation vapor pressure, but this measure together with a more effective WBF process might already be too efficiently remove clouds as it introduces a negative cloud cover bias. We additionally explored the sensitivity of low-level cloud cover to the strength of surface heat fluxes, and by increasing surface mixing, the observed cloud cover and cloud phase bias could also be reduced. As ECHAM6 already mixes too strongly in the Arctic regions, it is questionable if one can physically justify to increase mixing even further.

## 1 Introduction

With temperatures rising nearly twice as strongly compared to the temperature increase of the Northern Hemisphere (Screen and Simmonds, 2010), the Arctic reacts especially susceptibly to global climate change. This is due to several positive feedback mechanisms that strengthen the warming in the high latitudes (Serreze and Barry, 2011). This so-called Arctic Amplification

has important implications on the Arctic climate system like the extreme decrease in summer sea ice extent in recent years, the thawing of permafrost or the melting of glaciers in Greenland. Besides those effects on the regional scale, it is believed that Arctic Amplification might have effects on the atmospheric circulation due to a decrease in the temperature gradient between mid and high latitudes (Francis and Vavrus, 2012). Additionally, the melting glaciers in Greenland contribute to the sea level rise, which will affect many coastal areas around the globe.

While globally having a cooling effect, clouds in the Arctic warm the surface most of the year except a short period in summer (Intrieri, 2002; Zygmuntowska et al., 2012; Kay and L'Ecuyer, 2013). As the amount of clouds is thought to increase in a warming Arctic (Liu et al., 2012), their positive cloud radiative effect (CRE) can further enhance Arctic Amplification. Using global climate models (GCMs) to assess the CRE in the Arctic on a larger scale is inevitable because of the complexity of the climate system in the Arctic. Due to this complexity, even present day estimations of the CRE from climate models in the Arctic are inconclusive (Karlsson and Svensson, 2013), as those models still struggle to correctly simulate even basic properties like cloud cover and cloud distribution (English et al., 2015; Boeke and Taylor, 2016), which complicates an assessment of future Arctic warming. Another issue often found in GCMs is that they struggle to correctly simulate the phase state of clouds in the Arctic. As has been shown from ground-based (Shupe and Intrieri, 2004) and satellite observations (Cesana et al., 2012), liquid containing clouds are ubiquitous all over the Arctic and their CRE can significantly alter radiative budgets (Bennartz et al., 2013; Miller et al., 2015). Present-day climate models often underestimate the proportion of liquid to ice in mixed-phase clouds (Komurcu et al., 2014; Cesana et al., 2015; McCoy et al., 2016) which for some models this especially the case in the Arctic (Barton et al., 2012; Kay et al., 2016a). Correctly representing microphysical processes in Arctic mixed-phase clouds is key to correctly simulate typical features like their longevity (Morrison et al., 2011) and the typical two-state radiative structure of the Arctic boundary layer (Shupe and Intrieri, 2004; Stramler et al., 2011). As it has been shown by Pithan et al. (2014), models in which supercooled water freezes at too high temperatures often cannot reproduce the cloudy state. This consequently reflects on the radiative budget and temperature stratification as models that lack the cloudy state display excessive radiative cooling of the surface. One of the few GCMs that is able to reproduce the cloudy state in the Arctic is the MPI-ESM (Giorgetta et al., 2013). The MPI-ESM is consequently able to better simulate near surface stability compared to reanalyses (Pithan et al., 2014). This can be related to the fact the MPI-ESM is able to sustain liquid water in clouds even at relatively low temperatures in the polar regions (see Figure 5 and Figure 6 in Cesana et al., 2015). The presence of liquid water in the clouds also reflects on the net CRE of MPI-ESM as it exceeds the CMIP multi-model mean CRE by approximately $10\,\mathrm{W\,m^{-2}}$ and also is in good agreement with the CERES-EBAF net CRE (Boeke and Taylor, 2016). The existence of the cloudy state in the MPI-ESM also shows in the higher Arctic (low-level) cloud amount in this model, while most other models underestimate cloud amount (English et al., 2015).

As the MPI-ESM is quite different compared to other climate models when it comes to clouds in the Arctic, the main goal of this study is to identify processes and parametrizations that are responsible for the above mentioned features. To identify such processes, a thorough evaluation of the model using observations is necessary. Well suited for such an evaluation are datasets from satellite remote sensing. Satellites can provide observations on spatial and temporal scales much closer to the scales of GCMs and are therefore well suited for assessing the performance of such models. Satellite remote sensing in the

Arctic has to deal with several aspects that complicate their use in evaluating cloud properties in GCMs, which is especially the case for passive sensors. The polar night and often prevailing low-level inversions at high latitudes make it hard for passive instruments to discriminate between snow/sea ice and low-level clouds as they solely rely on the reflected and emitted radiation in the visible and thermal spectral ranges, respectively (Liu et al., 2010; Karlsson and Dybbroe, 2010). Active satellites like CloudSat (Stephens et al., 2002) and CALIPSO (Cloud-Aerosol Lidar and Infrared Pathfinder Satellite Observations; Winker et al., 2003) are better suited, as they are less affected by the environmental conditions in the Arctic than passive sensors (Zygmuntowska et al., 2012; Kay and L'Ecuyer, 2013). Additionally, active satellites can provide vertical profiles of cloud microphysical properties (especially CloudSat and to some extend also CALIPSO) which passive satellites can not provide. To facilitate the comparison of properties derived by satellites and the output from GCMs, the Cloud Feedback Model Intercomparison Project's (CFMIP) Observation Simulator Package (COSP; Bodas-Salcedo et al., 2011) has been developed. With the help of this satellite simulator, it is possible to consistently evaluate the results from GCMs by using common definitions of clouds observed from satellite and clouds simulated in GCMs. COSP has been used in various model evaluation studies (Nam and Quaas, 2012; Cesana and Chepfer, 2013; Nam et al., 2014), with some studies especially focusing on clouds in the Arctic (Barton et al., 2012; English et al., 2014; Kay et al., 2016a).

In the following, we will evaluate the performance of the atmospheric model ECHAM6 (Stevens et al., 2013), which is the atmospheric component of the MPI-ESM, in the Arctic and will especially focus on the representation of clouds in this remote region. We will compare COSP-derived output to the GCM-Oriented CALIPSO Cloud Product (GOCCP) dataset (Chepfer et al., 2010; Cesana and Chepfer, 2013), processed by the CFMIP Observations for Model Evaluation Project (CFMIP-OBS; Webb et al., 2017). Using this dataset ensures a consistent model-to-observation comparison as their diagnostics of observational data are consistent with the diagnostics within COSP. Based on the results of this evaluation, we conduct dedicated sensitivity studies that aim at identifying processes and parametrizations that could explain why ECHAM6/MPI-ESM is so different compared to other climate models in the Arctic.

## 2 Data and Model

### 2.1 ECHAM6 and COSP

In this study, we use the atmospheric model ECHAM6 (Stevens et al., 2013), developed by the MPI in Hamburg in its most recent version (ECHAM6.3; Mauritsen et al., 2019). In all our simulation, the model is run at a resolution of T63, which is equivalent to a Gaussian grid of approximately $1.875° \times 1.875°$ with 47 levels in the vertical. The model's vorticity and divergence are nudged to ERA-Interim reanalysis data (Dee et al., 2011) to enable comparison to satellite observations despite the relatively short run time of the model of less than 5 years. We use monthly observations of sea surface temperature and sea ice concentration from the AMIP II dataset (Taylor et al., 2000) as boundary conditions to further constrain the model.

To better compare the model results to the satellite observations, we use COSP (Bodas-Salcedo et al., 2011), version 1.4. Multiple satellite simulators are available within COSP, but here, only the simulator for CALIPSO (ActSim; Chepfer et al., 2008) is used. COSP uses model output like the profiles of temperature, pressure, cloud fraction, cloud water content, as well

as precipitation flux of rain and snow from large-scale/convective precipitation as input for its calculations. To enable a more consistent comparison between model and observed cloud properties, COSP divides each grid box into a specified number of subcolumns (here we use 40 subcolumns) to account for subgrid scale variability of grid-scale cloud properties (i.e. cloud cover and hydrometeors). For the subdivision of cloud properties into subcolumns, the Subgrid Cloud Overlap Profile Sampler (SCOPS) is used within the framework of COSP, that was originally developed as part of the ISCCP simulator (Klein and Jakob, 1999; Webb et al., 2001). It applies a pseudo-random sampling of cloud properties to be consistent with the cloud overlap assumption of the host model. Additionally, the precipitation fluxes in those newly created subcolumns are determined following a simple algorithm developed by Zhang et al. (2010). The calculations of the satellite simulators within COSP are then performed on each subcolumn to simulate specific signals received by the respective instrument and to mimic the retrievals derived from these instruments. By using the same instruments sensitivities and cloud overlap assumptions as used in GOCCP, COSP generates an output that is similar to the observations from satellites and also provides a common basis for comparing results from different climate models. The satellite simulator is implemented into ECHAM6 and is run online during the integration of the model. The output fields of COSP are interpolated onto the $2° \times 2°$ GOCCP grid for better comparison. For the evaluation of ECHAM6 in section 3, we run the model from 2007 to 2010, while for the sensitivity studies in section 4 we only run it for 2007 and 2008 to reduce computational cost.

## 2.2 GOCCP

To evaluate to what extent ECHAM6 is able to simulate cloud marco- (cloud cover) and microphysical (cloud phase) properties of Arctic clouds, we use the GOCCP dataset (Chepfer et al., 2010), which is generated from the CALIOP (Cloud-Aerosol Lidar with Orthogonal Polarization) Level 1B NASA Langley Atmospheric Sciences Data Center CALIPSO datasets. The CALIPO data in the GOCCP dataset is interpolated onto a $2° \times 2°$ grid in the horizontal and on a equally spaced vertical grid ($\Delta z = 480\,\mathrm{m}$) with 40 vertical levels ranging from the surface to $19\,\mathrm{km}$. On this grid, the lidar scattering ratio (SR) is computed by comparing the backscattered intensity of the lidar beam to that of a molecular atmosphere (no clouds or aerosols). A layer can then be classified as cloudy (SR > 5), clear (0.01 < SR < 1.2), fully attenuated (SR < 0.01) or unclassified (1.2 < SR < 5). Using these thresholds, cloud cover for different layers (low, mid, high) can be diagnosed. Those layers are defined as follows:

$$
\begin{array}{lll}
\text{high clouds} & p_{\text{top}} & < 440\,\mathrm{hPa} \\
\text{mid clouds} \quad 680\,\mathrm{hPa} > & p_{\text{top}} & \geq 440\,\mathrm{hPa} \\
\text{low clouds} & p_{\text{top}} & \geq 680\,\mathrm{hPa}
\end{array}
$$

Furthermore, the GOCCP dataset contains information on the phase of the cloud that is observed by CALIOP. By comparing the total backscattered lidar signal (ATB) to the perpendicularly (relative to the incident laser light) polarized backscattered lidar signal ($\mathrm{ATB}_\perp$), information on the shape of the particle that scattered the lidar beam can be retrieved. Assuming a scattering angle of 180°and no multiple scattering, a spherical particle does not change $\mathrm{ATB}_\perp$ while a nonspherical particle polarizes the backscattered lidar signal and consequently leads to a larger $\mathrm{ATB}_\perp$ (Cesana and Chepfer, 2013). Using a phase discrimination line that is a function of ATB and $\mathrm{ATB}_\perp$ (see Equation 3 in Cesana and Chepfer, 2013), one can distinguish in which phase state the scattering particle is. In late 2007, the nadir pointing angle of CALIPSO has changed to avoid spurious values of optical

properties in case of oriented crystals being present in clouds. As stated by Cesana et al. (2016), changing the nadir-pointing
angle resulted in less false cloud detection and less false liquid cloud determination since ice crystal plates produce the same
signature as liquid droplets. The effects of ex-/including the year 2007 are however rather small and can be attributed to internal
variability and do not affect our main conclusions in the following sections (see supplement) .

Even though an active sensor like CALIPSO is better suited for Arctic spaceborne remote sensing than passive sensors (Zygmuntowska et al., 2012; Kay and L'Ecuyer, 2013), it will also be affected by the atmospheric conditions at high latitudes, which
will introduce observational uncertainties. Due to the prevailing low-level, liquid containing clouds in the Arctic (Shupe and
Intrieri, 2004), the lidar beam can get attenuated by those optically thick clouds (Cesana et al., 2012). The lidar beam can not
penetrate through those low-level clouds and will cause an underestimation of clouds in the lowest layers of the atmosphere.
Comparing several CALIPSO-dervied datasets to ground based observations in Barrow and Eureka, Liu et al. (2017) showed
that near surface cloud cover can be underestimate by up to 40 % due to the attenuation of the lidar beam by those opaque,
low-level, liquid containing clouds. Even if the lidar beam is not attenuated and can reach down to the surface, clouds might
be missed by GOCCP. As Lacour et al. (2017) stated, using a SR > 5 to detect clouds can cause a significant underestimation of low-level ice clouds because those optically thin clouds with small vertical extent might be missed with such a high
detection threshold. Nevertheless, they found that the GOCCP dataset is superior over most passive spaceborne sensors as it
is much closer to ground based observations. Further uncertainty is introduced by different spatio-temporal sampling when
comparing ground based observation to spaceborne observation (Cesana et al., 2012; Liu et al., 2017). To circumvent some of
the reported issues, we not directly compare the modeled cloud cover to GOCCP but make use of COSP. By using the same
detection threshold for clouds, not suffering from similar attenuation effects of the (simulated) lidar beam and also comparing
the modeled and observed clouds on a similar spatial and temporal scale should enable a more consistent comparison.

To show that the COSP-derived cloud cover from ECHAM6 suffers from a similar underestimation of low-level cloud cover,
we compare modeled (ECHAM+COSP minus ECHAM) to observed (GOCCP minus ground based observations) cloud cover
profiles in Figure 1. For ground based observations, we use data from the 35-GHz millimeter cloud radars (MMCR) in Barrow
and Eureka as described in Shupe et al. (2011) for the period from 2007 to 2009. Similar to Liu et al. (2017), GOCCP underestimates the cloud amount in lowest levels of the troposphere by 15 to 20 % at both locations for reasons described above.
Looking at the difference between COSP- and ECHAM-derived (with that we mean cloud cover as diagnosed by the cloud
cover scheme in ECHAM6), we see that ECHAM+COSP also omits clouds close to the surface. Looking at the observed and
modeled differences of the cloud cover profiles, we find that the differences almost perfectly match for Barrow (except for
the lowest level which might be an artifact of vertically interpolating the data on the ECHAM6 grid). Differences at Eureka
also show an underestimation of cloud cover close to the surface, even if the difference of observed to modeled clouds does
not compare as well as for Barrow. Nevertheless, the comparison shown in Figure 1 make us confident that the observational
uncertainties present in the CALIPSO derived GOCCP dataset can in part be countered by using COSP derived cloud products,
which enables a fair comparison between observed and model clouds (Kay et al., 2016b).

## 3 Arctic clouds in ECHAM6

In the following, we evaluate the temporal mean of a nudged ECHAM6 run for the years spanning 2007 to 2011 with prescribed sea surface temperatures and sea ice concentration. For this comparison, we use monthly averaged GOCCP data for the same period that contain both daytime and nighttime overpasses. ECHAM6 + COSP is able to reproduce the general cloud amount and distribution as observed by GOCCP to some extent, but is biased high over the Arctic Ocean, Siberia and over the northern parts of Canada. Those areas correspond to areas that are covered with snow and sea ice, respectively. The overestimation of cloud cover in those areas is opposing the general low bias in cloud cover over the ocean and continental regions that are not covered by snow which might be due to the fact that ECHAM6 generally seems to simulate too few clouds at low and mid levels (Stevens et al., 2013).

To explore what causes the positive bias in cloud amount over snow and sea ice covered areas, it is important to know at which altitude the clouds are situated and of which thermodynamic phase (liquid or ice) they are composed. Figure 3 shows the meridional mean difference of ECHAM6 + COSP and CALISPO from 60°N to 82°N. Besides the difference in total cloud cover, Figure 3 also shows the difference in low, mid and high cloud cover (altitude bins defined as in subsection 2.2) as well as the difference in total liquid and total ice cloud cover. As low clouds are the most common cloud type in high latitudes, the difference in total cloud cover is strongly influenced by the difference of low-level clouds. For those low-level clouds, a clear influence of season and longitude on the difference in cloud cover can be observed, which is especially the case in winter and spring. During these two seasons and over nearly all regions (except the Atlantic Ocean), ECHAM6 + COSP simulates a greater cloud fraction than observed by GOCCP. As seen in Figure 2, there seems to be a connection between the snow/sea ice coverage of the surface which can also be observed in Figure 3. Besides low-level clouds, high-level clouds also seem to be not simulated correctly in ECHAM6. The model generally overestimates the amount of high-level clouds, but in contrast to low-level clouds, they do not really show a dependency on longitude and only a weak dependency on the season. For mid-level clouds, cloud cover almost perfectly matches the observations in spring and fall, whereas in summer/winter, mid-level cloud cover is underestimated/overestimated by the model. For spring, summer and fall no significant dependency on longitude is distinguishable which is not the case for winter where a similar can be observed as for low-level clouds. The reason for seasonal variation of mid-level clouds is caused by the varying height of the troposphere, which is dependent on the tropospheric temperature profile. For colder temperatures, the tropopause is much lower than for warmer temperatures which causes cirrus clouds to vary in altitude. Therefore, some of the cirrus clouds in ECHAM6 are considered mid-level clouds in winter which is not the case for GOCCP. This effect reveres in summer, when ECHAM6 underestimates the amount of mid-level clouds when ECHAM6 simulates the bulk of the cirrus clouds at higher altitudes. When further discriminating between ice- and liquid-containing clouds (bottom row in Figure 3), one finds that this seasonal variation with a too large cloud cover in winter and spring mainly stems from an overestimation of liquid-containing clouds that usually can be found in the lower troposphere. In the Arctic, liquid containing clouds are of special importance as those clouds strongly influence the radiative budget at the surface due to their large optical thickness and strong effect on net surface longwave radiation (Shupe and Intrieri, 2004) which causes a warming at the surface. For ice clouds, on the other hand, only very little seasonal

or longitudinal variability in the deviation is distinguishable, and it is comparable to the difference in high cloud cover as those high clouds mainly consist of ice particles. Taken together, ECHAM6 simulates low-level, liquid containing clouds too frequently, and this overestimation appears to be connected to properties of the underlying surface. Additionally, high-level clouds are also overestimated, but this should not be subject of this study. We additionally performed a comparison of modeled cloud fraction profiles to ground based profiles from two cloud radars (see supplement). The comparison shows that compared to ground-based observations, the model slightly overestimates cloud fraction in layers close to the surface, even though not as pronounced as it is the case for the ECHAM6+COSP / CALIPSO-GOCCP difference. Such a comparison nevertheless has to be interpreted with care as spatial scales of modeled and observed quantities do not match and also due to fundamental differences in the way physical properties are diagnosed in the model and in observations.

The cloud cover and moisture bias (see Appendix A) implies that either the removal of atmospheric moisture by precipitation or fluxes of moisture from the surface into the atmosphere are not represented correctly in the model and that this seems to be connected to the underlying surface. Moisture fluxes into the atmosphere are directly influenced by surface properties like surface roughness (which can be reduced by snow on the surface) or availability of humidity at the surface (which itself is a function of temperature) and indirectly through increased stability of the layers close to the surface that consequently has an influence on vertical mixing of momentum and latent/sensible heat fluxes. The linkage between surface properties and moisture removal can be established through the modification of the atmospheric stratification as the strong radiative cooling causes the temperatures to be significantly lower compared to a snow- and ice-free surface. Possibly, temperature dependent processes like the Wegener-Bergeron-Findeisen process (Wegener, 1911; Bergeron, 1935; Findeisen, 1938) or the heterogeneous freezing might not sufficiently turn liquid water into ice in those regions, which we will investigate in the following section.

## 4   Sensitivity studies

In this section, we will examine how sensitively cloud cover and cloud phase react to modifications of cloud microphysical parametrization and to modified surface fluxes of latent/sensible heat. As we have shown in the previous section, it is mainly the low-level, liquid containing clouds that cause the low clouds bias in ECAHM6. Low-level clouds in the Arctic are typically mixed-phase clouds, so the overestimation of liquid clouds can be related to a misrepresentation of microphysical processes that act in this temperature regime, i.e., heterogeneous freezing of cloud liquid into ice or the production of cloud ice at the expense of cloud liquid water, also known as the Wegener-Bergeron-Findeisen (WBF) process. As most precipitation in higher latitudes is formed by the aforementioned process, a higher ice content should lead to the dissipation of clouds, as can be seen in the rather rapid transition from the cloudy into the clear state that is often observed in the Arctic (Morrison et al., 2011). Previously, Klaus et al. (2012) explored the sensitivity of cloud microphysical properties in a single column setup of the regional Arctic climate model HIRHAM5, which also uses the physical parametrizations of ECHAM. They modified several commonly used microphysical tuning parameters and only a stronger WBF process and a more effective collection of cloud droplets by snow were able to reduce the liquid water content. Additionally, we conducted a sensitivity study to explore the effect of an increased efficiency of heterogeneous freezing of cloud droplets, which also reduced the liquid water content. Out

of the three processes, the WBF process was by far the most efficient in turning cloud liquid into cloud ice and was also used
by Klaus et al. (2016) to tune the microphysics in HIRHAM5, who reported a similar overestimated amount of liquid clouds.
In our study, we will therefore explore the effect of different strengths of the WBF process on cloud cover and cloud phase.
Depositional growth of cloud ice takes place, according to the ECHAM6 parameterizations, if one of the following conditions
is met:

1. $T < -35°\,\mathrm{C}$

2. $T < 0°\,\mathrm{C}$ and $x_i > \gamma_{\mathrm{thr}}$ (where $x_i$ is the in-cloud ice mixing ratio)

The second conditions can be seen as a simple parametrization of the WBF process, as it allows deposition/condensation of
ice/liquid to take place for temperatures below $0°\,\mathrm{C}$ if the ice mixing ratio within the cloud is above/below a certain value.
In ECHAM6 and other climate models, the WBF process is often strongly simplified. As can be seen from the condition for
the onset of the WBF process in ECHAM6, there is no explicit dependence of this process on vertical velocity. Korolev and
Mazin (2003) have shown that only if the updraft speed $u_z$ within a cloud is less than a threshold vertical velocity $u_z^*$, the WBF
process can deplete any excess water vapor at the expense of liquid water within the cloud. $u_z^*$ is defined as follows:

$$u_z^* = \frac{e_s - e_i}{e_i}\, \eta\, N_i\, \overline{r}_i \tag{1}$$

where $e_s/e_i$ is the saturation vapor pressure over liquid/ice, $\eta$ a coefficient dependent on temperature and pressure, $N_i$ the ice
crystal number concentration and $\overline{r}_i$ the mean radius of the ice crystals. Assuming $\frac{e_s - e_i}{\eta}$ to be constant, $u_z^*$ and therefore the
condition for the onset of the WBF process (for a given $u_z$ and a given temperature) is only function of $N_i\, \overline{r}_i$. As ECHAM6
uses a single moment microphysical scheme, only information on the ice mixing ratio is present. As the ice mixing ratio also
can be calculated as a function of $N_i$ and $\overline{r}_i$ might at least partly justify the use of $\gamma_{\mathrm{thr}}$ as a threshold for the onset of the WBF.
Nevertheless, this is quite an strong simplification for the onset of this process as it is now independent on vertical velocity.
This also reflects on the fact that $\gamma_{\mathrm{thr}}$ is resolution-dependent in ECHAM6 and can vary by an order of magnitude between the
different horizontal resolutions of ECHAM6.

Due to this strong variation of $\gamma_{\mathrm{thr}}$ for different horizontal resolutions and due to the fact that it is one of the few parameters that
is able to reduce the liquid water content of clouds in the Arctic (Klaus et al., 2012), we will now explore how sensitive cloud
cover and cloud phase reacts to change in $\gamma_{\mathrm{thr}}$. Lower values of $\gamma_{\mathrm{thr}}$ increase the effectiveness of the WBF process, leading to
less cloud water but more cloud ice to be present. As almost all precipitation in the Arctic is formed via the ice phase, a decrease
of $\gamma_{\mathrm{thr}}$ is expected to eventually lead to a decrease in cloud cover as cloud condensate should be more efficiently removed via
precipitation. As can be seen from Figure 4, decreasing $\gamma_{\mathrm{thr}}$ in fact leads to a reduction in low-level liquid-phased clouds in
winter. It also can be seen that liquid cloud fraction decreases quite strongly if one halves the $\gamma_{\mathrm{thr}}$ and that this decrease is more
effective over continental regions compared to oceanic regions. Despite this fact, tuning low-level liquid cloud cover to match
the observed liquid cloud cover of GOCCP using the WBF process alone poses difficulties. Setting $\gamma_{\mathrm{thr}}$ to $2.5 \cdot 10^{-6}\ \mathrm{kg\,m^{-3}}$
or lower improves low-level liquid cloud cover east of $90°$ E, but introduces and further strengthens an already observable low
bias in low-level, liquid clouds between $315°$ E and $90°$ E in ECHAM6. This implies that tuning the WBF can not be used to

tune the cloud microphysics alone. Due to the fact that other processes that are able to reduce the liquid water content (more effective collection of cloud droplets by snow and heterogeneous freezing) do not do this in a sufficiently strong manor, we nevertheless think that increasing the efficiency of the WBF process is the most promising approach to tune Arctic cloud phase.

In the evaluation of cloud phase in section 3, the cloud phase ratio is used, which only can provide information of cloud phase as long as the lidar beam is not attenuated. This might cause some clouds to be missed in GOCCP and also in COSP, especially if clouds contain water. Therefore, we will look at the mass phase ratio as it is simulated by the model directly so that phase ratio is not affected by the attenuation of the lidar beam. To estimate how ice mass fraction is simulated in ECHAM6, we look at temperature-binned ice fraction in the North Atlantic and Siberia and how ice fraction changes for lower values of $\gamma_{thr}$ in

Figure 5. For the North Atlantic, clouds mostly consist of ice up to a temperature of $-10°$ in the default setting of $\gamma_{thr}$ before clouds start to become more liquid. The ice fraction in Siberia already decreases at colder temperature and then stays more or less constant at a value of 0.7 up until $-5°$. Comparing this to in-situ observation of ice fraction as provided by Korolev et al. (2017) such a "plateau" is not visible. Figure 5-14 in Korolev et al. (2017) shows a more gradual increase in ice fraction with decreasing temperature (which can be seen in the bins for high/low ice fraction) and we think that the more or less constant

ice fraction in the model over Siberia is another indication of an overestimated amount of liquid clouds over snow/ice covered surface as has been shown in Figure 3. As the ice fractions from in-situ observations and the ice fractions from the model are on a completely different spatial scale, one nevertheless has to be careful when doing such a comparison. To our knowledge, there is no observational product available that can provide liquid water and ice water content on a global scale. A possible approach to evaluate cloud phase would be to look at liquid/ice water path which can be derived from MODIS. As stated in the intro-

duction, using passive spaceborne sensors might be problematic due to the environmental conditions and also due to fact the Arctic clouds are often mixed-phase clouds, which further complicates the retrieval of cloud microphysical properties (Khanal and Wang, 2018). Decreasing $\gamma_{thr}$ has quite a strong effect on the ice fraction over Siberia where ice fraction is increased and the general shape of the curves over the North Atlantic and over Siberia are now quite similiar to each other. While a higher value of $\gamma_{thr}$ might be able to remedy the bias of liquid cloud over snow and ice covered surfaces, a too high value of $\gamma_{thr}$ will

lead to an underestimation of liquid clouds over open water. As liquid clouds react rather sensitively to a more effective WBF process, only minor changes of $\gamma_{thr}$ can have strong effects on the amount of liquid clouds and we think that setting $\gamma_{thr}$ to $2.5 \cdot 10^{-6}$ kg m$^{-3}$ is the best choice to revise the WBF process. This value is a good compromise between improving cloud cover/phase over snow and ice covered surfaces by simultaneously not further worsen clouds in other regions.

Even though a more effective WBF is able to reduce low-level liquid cloud cover, the overall low-level cloud cover remains

more or less unchanged. This is striking, as one would expect cloud cover to decrease due the stronger removal of cloud condensate by precipitation in ice clouds. A possible explanation why changing the strength of the WBF process does not result in a significant change in cloud cover is the way saturation water vapor pressure is calculated in the cloud cover scheme. For temperatures below $0°$ C, the saturation water vapor pressure in ECHAM6 can either be calculated with respect to water or ice. As saturation water vapor pressure over ice decreases faster with decreasing temperature compared to the saturation water

vapor pressure over water, relative humidity with respect to ice will be larger compared to relative humidity with respect to water at the same water vapor pressure at sub-zero temperatures. For the decision with respect to which phase state the satura-

tion water vapor is calculated, ECHAM6 uses the same conditions as for the WBF process, so if depositional (condensational) growth of ice crystals (cloud droplets) takes place, saturation water vapor pressure is calculated with respect to ice (water). As cloud cover is diagnosed as a function of grid-mean relative humidity (Sundqvist et al., 1989), the choice with respect to which

phase state the saturation water vapor pressure is calculated has a significant effect on fractional cloud cover. For the same water vapor pressure, relative humidity and therefore cloud cover will be much higher if cloud ice content exceeds $\gamma_{\mathrm{thr}}$. This explains why enhancing the efficiency of the WBF process by choosing lower values for $\gamma_{\mathrm{thr}}$ has only a minor effect on cloud cover. As one decreases $\gamma_{\mathrm{thr}}$, saturation water vapor pressure is more frequently calculated with respect to ice, which allows clouds to form at lower water vapor contents. Furthermore, as an existing liquid cloud starts glaciating, in this parameterization

the cloud cover will increase instantaneously once the ice content exceeds the threshold. As the Sundqvist cloud cover scheme is not able to handle supersaturation with respect to ice, a grid box is also often completely cloud covered at sufficiently low temperatures (Lohmann et al., 2008; Bock and Burkhardt, 2016).

To avoid this sudden increase in cloud cover as soon as the ice water content becomes greater than $\gamma_{\mathrm{thr}}$, we modified the calculation of the saturation water vapor pressures in the cloud cover scheme by using a weighted average between the saturation

water vapor pressures over liquid water, $e_l$, and ice, $e_i$:

$$e = e_l(1 - f_i) + e_i f_i. \tag{2}$$

$f_i$ is a weighting factor where $f_i = 0$ for a water cloud, $f_i = 1$ for an ice cloud and $0 < f_i < 1$ for a mixed-phase cloud (Korolev and Isaac, 2006). One commonly used approach to determine $f_i$ is to define it as a temperature-dependent function that aims to resemble the partitioning between cloud water and cloud ice with decreasing temperatures (Fowler et al., 1996; Morrison and

Gettelman, 2008; Dietlicher et al., 2018a). We use a linear function that interpolates between the melting point $T_{\mathrm{ice,1}} = 0°\,\mathrm{C}$ and the homogeneous freezing threshold $T_{\mathrm{ice,2}} = -35°\,\mathrm{C}$ and define $f_i$ as follows:

$$f_i = 1 - \frac{T - T_{\mathrm{ice,2}}}{T_{\mathrm{ice,1}} - T_{\mathrm{ice,2}}}. \tag{3}$$

$f_i$ is set to 1 for temperatures lower than $-35°\,\mathrm{C}$, while for $T > 0°\,\mathrm{C}$, $f_i$ is fixed to 0. In case the cloud ice content is less than $\gamma_{\mathrm{thr}}$, we also set $f_i$ to 0. This condition is used to delay cloud formation as long as there is not enough cloud ice for the WBF

process to efficiently produce cloud ice and the phase of the clouds is predominantly liquid. Compared to the previous way of defining the saturation water vapor, this new approach introduces supersaturation with respect to ice of up to 10% for clouds in the temperature regime of mixed-phase clouds.

In Figure 6, we compare the effects of this new saturation water vapor pressure calculation (NEW) to the standard calculation for low-level cloud cover (BASE) in DJF for different settings of $\gamma_{\mathrm{thr}}$. As it also was found in Figure 4, Arctic low-level

cloud fraction bias remains more or less unchanged in the BASE runs for a more efficient WBF process. The reduction of the liquid-cloud bias due to a more effective WBF is almost completely compensated by an increased positive bias in low-level ice clouds. This increase in low-level ice clouds can be attributed to the fact that the ice water content becomes greater than $\gamma_{\mathrm{thr}}$ and the saturation water vapor pressure is more frequently calculated with respect to ice. This enables clouds to be present even at lower value of absolute humidity compared to higher values of $\gamma_{\mathrm{thr}}$. Compared to standard way of calculating saturation

water vapor pressure, the temperature weighted scheme is able to keep the amount of ice clouds unchanged while decreasing the amount of liquid clouds. As the amount of low-level ice clouds remains more or less unchanged with this newly introduced scheme, the loss in cloud cover correlates with the loss in liquid clouds due to the more effective WBF process. As stated above, tuning the WBF process alone was not able to completely remedy the overestimated amount of low-level, liquid clouds over snow and ice covered regions and additionally introduced a negative bias over oceanic regions. This explains why even with

this newly introduced way of calculating saturation water vapor pressure in the cloud cover scheme, it is difficult to globally improve the amount of low-level clouds.

As we have shown in the section above, it is difficult to tune cloud cover and phase using cloud microphysical parameterizations. As the cloud bias in ECHAM6 seems to be related to snow and ice covered surfaces, it is possible that fluxes of moisture from the surface into the atmosphere are not represented correctly in the model. In ECHAM6, turbulent surface fluxes of either

heat ($\psi = h$) or momentum ($\psi = m$) are described using the following bulk-exchange formula:

$$\overline{w'\psi'} = -C_\psi \, |\boldsymbol{V}| \, (\psi_{\mathrm{nlev}} - \psi_{\mathrm{sfc}}), \tag{4}$$

where $C_\psi$ is the bulk exchange coefficient with respect to $\psi$, $|\boldsymbol{V}|$ is the difference of the absolute wind velocity at the surface and the wind velocity in the lowest model level and the last term in parentheses is the difference of the respective quantity between the first model level ($\psi_{\mathrm{nlev}}$) and at the surface ($\psi_{\mathrm{sfc}}$). $C_\psi$ can be further separated into the product of a neutral limit

transfer coefficient $C_{\mathrm{N},\psi}$ (which only depends on surface properties like surface roughness and the height of the first model level) and a (surface-layer) stability function $f_\psi$:

$$C_\psi = C_{\mathrm{N},\psi} \, f_\psi \tag{5}$$

Those stability functions can be derived from Monin-Obukhov similarity theory by integrating the flux-profile relationships from the surface up to the lowest model layer but this is not practical for climate models. Therefore, ECHAM6 uses empirical

expressions for those stability functions similar to the ones proposed by Louis (1979), depending on both surface properties and stability of the layer between the surface and the lowest model level (expressed by the moist Richardson number). To obtain a first impression on how cloud cover reacts to increased/decreased surface fluxes, we introduced a scaling factor $\mu$ into Equation 5 so that it becomes:

$$C_\psi = \mu \, C_{\mathrm{N},\psi} \, f_\psi. \tag{6}$$

This scaling factor can be used to increase or decrease the neutral limit transfer coefficient which can be interpreted as a modification of the surface roughness, where values of $\mu$ greater than 1 denote higher surface roughness and stronger mixing, while values of $\mu$ less than 1 denote lower surface roughness and reduced mixing, respectively. We only modify this scaling factor for snow and sea ice covered surfaces and set it to 1 elsewhere. As before, a surface is considered snow-covered if snow height is higher than an arbitrarily chosen value of 2 cm and, a surface is considered sea ice covered if more than 50 % of a grid

box is covered by sea ice. In Figure 7 we show the effect of increasing ($\mu = 5$) and decreasing ($\mu = 0.2$) mixing on low-level cloud cover over those surfaces in the northern hemisphere (for comparison we also added GOCCP cloud cover). For sea ice

covered surfaces, increased mixing ($\mu = 5$) leads to reduced low-level cloud cover during winter and spring, while in summer, it leads to an increase in cloud cover compared to base run ($\mu = 1$). For decreased mixing ($\mu = 0.2$), exactly the opposite is simulated, with more clouds in winter and fewer clouds during summer compared to the basic setup. Total cloud cover behaves

similarly for increased/decreased mixing whenever a grid box is snow covered (no information is available during summer as no grid box is snow-covered). If one further discriminates between liquid and ice clouds, the effect of decreasing/increasing surface fluxes mainly shows for low-level liquid clouds while the amount of low-level ice clouds remains more or less unchanged. By increasing surface fluxes by a factor of 5, the positive bias of liquid clouds in winter vanishes and almost perfectly matches the lidar-derived cloud mount except for fall this measure leads to an underestimated cloud amount.

In general, increased mixing is expected to increase the moisture fluxes from the surface into the atmosphere and therefore to increase the moisture availability in the lowest levels of the atmosphere. While this assumption is valid for most parts of the globe, heat fluxes in the Arctic can reverse during winter so that fluxes of sensible and latent heat from the lowest layers of the atmosphere are directed towards the surface. This is due to the often observed low-level temperature inversions that also lead to qualitatively similar moisture profiles as saturation water vapor content is a function of temperature. In case of such a

moisture inversion, increased mixing increases the latent heat fluxes from the atmosphere onto the surface, and this process is a sink for atmospheric moisture. In case of a temperature inversion, stronger mixing causes surface temperatures to increase, but the effect of this temperature increase on cloud cover is twofold. On the one hand, warmer surface temperatures make the atmospheric stratification less stable, which further increases mixing and consequently leads to stronger removal of atmospheric moisture by latent heat fluxes as long as the moisture inversion is still present. On the other hand, a warmer surface increases

the moisture content. Consequently, the vertical moisture gradient is weakened, also resulting in weaker moisture fluxes from the atmosphere onto the surface according to Equation 4. Altogether, the increased moisture removal seems to dominate over the decrease in vertical moisture gradient, as cloud cover is reduced due to stronger mixing. Despite the potential to improve cloud cover by stronger surface mixing over snow and ice covered surfaces, it is questionable whether one can physically justify to further increase mixing as most climate models already mix too strongly in stable boundary layers (Holtslag et al.,

2013). We will further elaborate on that in the next section.

## 5    Discussion

In the previous sections, we showed that ECHAM6 overestimates low-level cloud cover over snow- and ice-covered surfaces during wintertime compared to the GOCCP dataset. To this end, we conducted sensitivity studies to explore the effect on clouds in ECHAM6 by varying the efficiency of several physical processes. While the partitioning of liquid and ice clouds

can be improved by a more effective WBF process, the overall positive cloud cover bias could not be reduced by that measure alone. We showed that this positive cloud cover bias can be improved by an alternative approach of calculating the saturation water vapor pressure in the cloud cover scheme. Nevertheless, it is questionable to what extend a more effective WBF process in ECHAM6 can be used to improve Arctic cloud properties. Besides the effect of cloud microphysics on cloud cover, we additionally explored the effect of stronger/weaker surface mixing on cloud cover and showed that increased mixing in ECHAM6

leads to a reduction of low-level clouds and by reducing liquid clouds. We will now discuss whether the two approaches can be used to tune Arctic cloud cover and cloud phase in ECHAM6.

As climate models in general struggle to represent microphysical processes correctly, attributing the positive bias in cloud cover to misrepresented microphysical processes seems not to be far-fetched. We explored the sensitivity of cloud cover to changes in the effectiveness of the WBF process and showed that it can be used to reduce liquid cloud cover in ECHAM6. Additionally, this measure is slightly more effective over snow- and ice-covered surfaces which helps to reduce the positive bias in liquid clouds in those regions. Unfortunately, increasing the effectiveness of the WBF process alone also introduced a negative bias over oceanic regions. This hints that just revising the effectiveness of this process alone might not be sufficient to improve cloud phase on global scale. We also showed that the way microphysical processes act is not straightforward, as one might expect a higher removal of atmospheric moisture for a higher cloud ice content that should eventually decrease cloud cover. As it seems impossible to reduce cloud cover in ECHAM6 through microphysics alone, we switched to a different approach for calculating saturation water vapor pressure in the cloud cover scheme. By using a temperature-dependent linear function that interpolates between saturation with respect to water and saturation with respect to ice, we were able to reduce cloud cover in the temperature range of typical mixed-phase clouds. Previously, the decision with respect to which phase the saturation water vapor pressure is calculated was primarily based on a cloud ice threshold to be consistent with parametrization of the WBF within the microphysical scheme. For the WBF process, such a threshold is an appropriate choice as we discussed above, but when used in the cloud cover parameterization it might introduce spurious increases in cloud cover when prexisting liquid clouds start to glaciate. By using a new temperature dependent calculation of the saturation water vapor pressure, we allowed for a slight supersaturation with respect to ice in the cloud cover scheme so that relative humidity was reduced when diagnosing cloud cover using the Sundqvist scheme. Allowing for supersaturation with respect to ice is crucial to accurately represent mixed-phase and ice clouds as supersaturation with respect to ice is frequently observed in clouds that contain ice (Heymsfield et al., 1998; Gierens et al., 2000; Spichtinger et al., 2003; Korolev and Isaac, 2006). As discussed in Dietlicher et al. (2018b), calculating the saturation water vapor pressure as a function of temperature alone might not be an appropriate choice as it does not arise from a valid solution of the Clausius-Clapeyron equation. Besides the positive effect of properly accounting for supersaturation with respect to ice in the mixed-phase temperature regime, it might also be beneficial for the simulation of cloud cover below the homogeneous freezing threshold. Even with the revised calculation of saturation water vapor pressure, ice clouds are still slightly overestimated in the Arctic (see Figure 6). This, together with the fact that ECHAM6 largely overestimates cirrus cloud emphasizes the need for a cloud cover parametrization that is designed to handle supersaturation with respect to ice even at temperatures below the homogeneous freezing threshold. First attempts to implement such a parametrization were made by Bock and Burkhardt (2016) and Dietlicher et al. (2018b) for ECHAM-HAM, that uses a more sophisticated two-moment microphysics scheme that explicitly allows ice supersaturation (Lohmann et al., 2008). Even though their revised cloud cover schemes were primarily intended to improve cirrus clouds, it is to be expected that such an approach might also improve low-level cloud cover in the Arctic as those clouds often contain ice even though those schemes can not be implemented into ECHAM6 due to the simpler single-moment microphysics. Klaus et al. (2016) used a different approach to reduce Arctic cloud cover for their regional Arctic climate model HIRHAM5 (same physical parametrizations as ECHAM6

but different dynamical core). Instead of using the diagnostic Sundqvist scheme with its uniform probability density function, they used the statistical Tompkins (2002) cloud cover scheme and modified the shape of the beta function that is used as the probability density function to diagnose cloud cover. By making the beta function negatively skewed, they were able to reduce the positive cloud cover bias in their model. The Tompkins (2002) cloud cover scheme is presently not available in ECHAM6 which prevents us from evaluating their approach on a more global scale.

Besides attributing the positive bias in cloud cover to misrepresented microphysical processes, we additionally focused on the effect of surface fluxes on Arctic clouds in ECHAM6. By increasing the surface mixing, we were able to improve both the biases in cloud cover and cloud phase. As we have already stated in the previous section, further increasing mixing over snow and ice covered regions might not be desirable as climate models in general mix too strongly under these conditions (Davy and Esau, 2014). That this is also the case for ECHAM6 can be confirmed by two different aspects within the parametrization of the

surface mixing in ECHAM6. In the following, we only discuss mixing over sea ice, but the conclusions are to some extent also valid for snow covered surfaces. From Equation 5, we see that the bulk exchange coefficient that governs the strength of mixing in ECHAM6 is calculated as the product of the neutral limit transfer coefficient $C_{N,\psi}$ and a (surface-layer) stability function $f_{\psi}$. The roughness length for both momentum and scalars is set to $z_{0,h/m} = 10^{-3}$ m over sea ice, which is rather large compared to observations. Citing several observational studies, Gryanik and Lüpkes (2018) stated that roughness length for momentum

over ice covered surface can have values ranging between $z_{0,m} = 7 \cdot 10^{-6}$ m and $z_{0,m} = 5 \cdot 10^{-2}$ m with an average value of $z_{0,m} = 3.3 \cdot 10^{-4}$ m (Castellani et al., 2014), but surface roughness can locally be enhanced way beyond the values given by Gryanik and Lüpkes (2018), e.g. in the marginal sea ice zones or at large sea ice ridges in the central Arctic or near Greenland (Lüpkes et al., 2012). The average value is already an order of magnitude lower then the roughness length used in ECHAM6, so neutral limit transfer coefficients are also larger than the observations suggest. The same is true for the stability function $f_{\psi}$

over sea ice in stable regimes. Gryanik and Lüpkes (2018) compared the stability functions used in ECHAM6 (Louis, 1979) to an alternative formulation of those functions that were derived from the SHEBA dataset (Grachev et al., 2007) that should be better suited for stable stratification over sea ice. While for weaker stability, the presently used stability functions are in agreement with this new formulation, they are considerably larger for stronger stability. As both the presently used roughness length over ice covered surface and the stability functions applied in ECHAM6 already produce stronger mixing than observed,

it is questionable if one can physically justify to even further increase surface mixing over snow- and ice-covered surfaces.

## 6  Conclusions

In this study, we explored potential causes for the overestimated cloud cover in ECHAM6 and identified two physical processes - cloud microphysics and surface fluxes - that might be responsible for this. Especially mixed-phased clouds pose a challenge for climate models, as many of the processes acting in mixed-phase clouds are only poorly understood, which makes it even

harder to develop cloud microphysical parametrization. As we have shown, ECHAM6 also struggles to correctly simulate mixed-phase clouds which might be attributed to the oversimplified representation of the WBF processes. However, simply increasing the efficiency of the WBF process leads to almost completely glaciated clouds below $0^{\circ}$ C and thus introduces a bias

that is also found in several other climate models. Additionally, it would be beneficial to revise the cloud cover scheme as it presently is not able to handle supersaturation with respect to ice. We also explored the sensitivity of cloud cover to modified surface fluxes and showed that is possible to reduce the cloud cover bias in ECHAM6 through stronger surface mixing. As stated above, increasing surface mixing even further might not be desirable in ECHAM6 but the opposite approach can be used to improve the representation of clouds in other climate models, as many of them underestimate Arctic cloud cover. Altogether, this study provides valuable information on possible reason why ECHAM6/MPI-ESM is so different with respect to clouds compared to other models and lessons learned from this study can be beneficial for other models when it comes to representation of clouds in the Arctic.

## Appendix A:  Arctic relative humdity bias in ECHAM6

To show that the above reported overestimated amount of low-level clouds is not just due to possible observational uncertainties in the GOCCP, we additionally assess how well the model is able to reproduce profiles of temperature and relative humidity in the Arctic. Cloud cover in ECHAM6 is diagnosed as a function of grid-scale relative humidity (see Equation 3). At least from the model side, high values of relative humidity are indicative of a high cloud cover. We therefore compare profiles of temperature and humidity from the model to profiles measured by radiosondes within high latitudes. Additionally, we used data from ERA-Interim (Dee et al., 2011) to obtain further information about the stratification besides the spatially limited profiles from radiosondes. Due to the sparse availability of observational data in high latitude, one should not take data from ERA-Interim at face value, but it provides nevertheless another estimate to evaluate ECHAM6. To make the profiles of the various stations independent of surface elevation, we use height above the ground as the vertical coordinate in our analysis and linearly interpolate the radiosonde data to altitudes above the surface spanning from 0 m to 1000 m in steps of 500 m. Using such a vertical coordinate facilitates the comparison of several stations that might vary in surface elevation. Additionally, it is independent of synoptic situation which would not be the case if one uses pressure as the vertical coordinate. A disadvantage of this vertical coordinate is that the surface elevation in the model and the reanalysis is a grid-box mean which can deviate from the actual surface elevation of the station. As most stations are situated near the coast or within the rather flat plains of the Siberian tundra, we expect only minor inconsistencies. One also has to keep in mind that the vertical resolution of the soundings, ECHAM6 and ERA-Interim is rather poor, so only a certain level of detail can be expected from them. Figure A1 shows that ECHAM6 underestimates surface temperature compared to ERA-Interim in large parts of high latitudes. In contrast, the difference of ECHAM6 to radiosonde profiles shows a slight positive bias, especially over Siberia. This discrepancy between ERA-Interim and the radiosondes is not as large at 500 m and 1000 m AGL. At those altitudes, ECHAM6 is in good agreement with the observations and ERA-Interim. Looking at the biases in relative humidity, both ERA-Interim and the radiosonde profiles show that ECHAM6 seems to overestimate relative humidity at the surface. This overestimation is most strongly pronounced over Siberia and northern America which is consistent with the overestimated low-level cloud cover in those regions as shown in Figure 3. Even though a direct relationship between cloud cover and relative humidity should not be interpreted as a watertight evidence, the positive bias in relative humidity (compared to reanalysis and

radiosondes) supports our initial claim of an overestimated cloud fraction in high latitudes in ECHAM6 as we have shown using satellite observations.ppendix

*Author contributions.* JK and JQ conceived this study. JM and MS helped to set up COSP in ECHAM6 and helped in conducting the model runs. MS further contributed by providing valuable expertise on the physical parametrizations of ECHAM6. All of the authors assisted with
the interpretation of the results. JK prepared the manuscript with contributions from all co-authors.

*Competing interests.* The authors declare that they have no conflict of interest.

*Disclaimer.* TEXT

*Acknowledgements.* We gratefully acknowledge the funding by the Deutsche Forschungsgemeinschaft (DFG, German Research Foundation) - Projektnummer 268020496 - TRR 172, within the Transregional Collaborative Research Center "ArctiC Amplification: Climate Relevant
Atmospheric and SurfaCe Processes, and Feedback Mechanisms (AC)[3]. The ECHAM6 model is developed by the Max Planck Institute for Meteorology, Hamburg, and we thank the colleagues for making the model available to the research community. Simulations were conducted at the German Climate Computing Centre (Deutsches Klimarechenzentrum, DKRZ). We would like to thank NASA and CNES for operating the CALIPSO satellite, as well as the data producers for the satellite data used in this study.

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

690

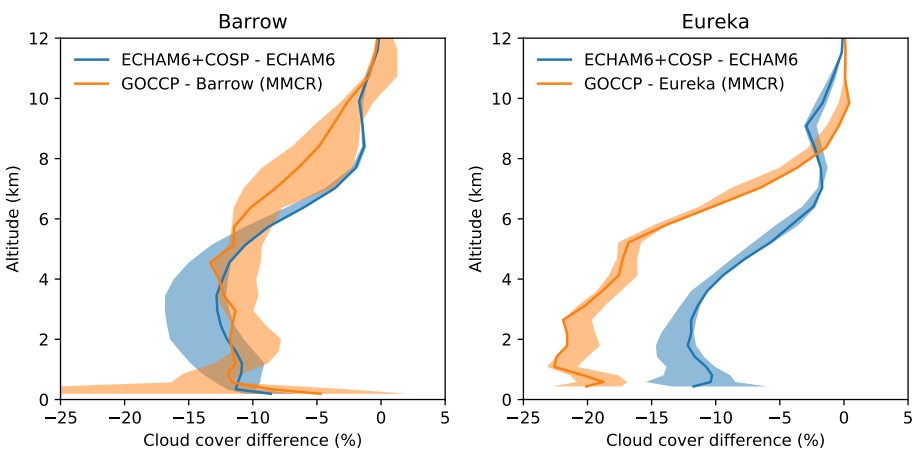

**Figure 1.** Difference in cloud cover profiles (from 2007 to 2009) of ECHAM6+COSP minus ECHAM6 and GOCCP minus ground based observations. Cloud cover profiles from ground based observations are derived from 35-GHz millimeter cloud radars (MMCR) in Barrow and Eureka as described in Shupe et al. (2011). Shaded areas show the effect of using the neighboring gridpoints around the location in in the grided data.

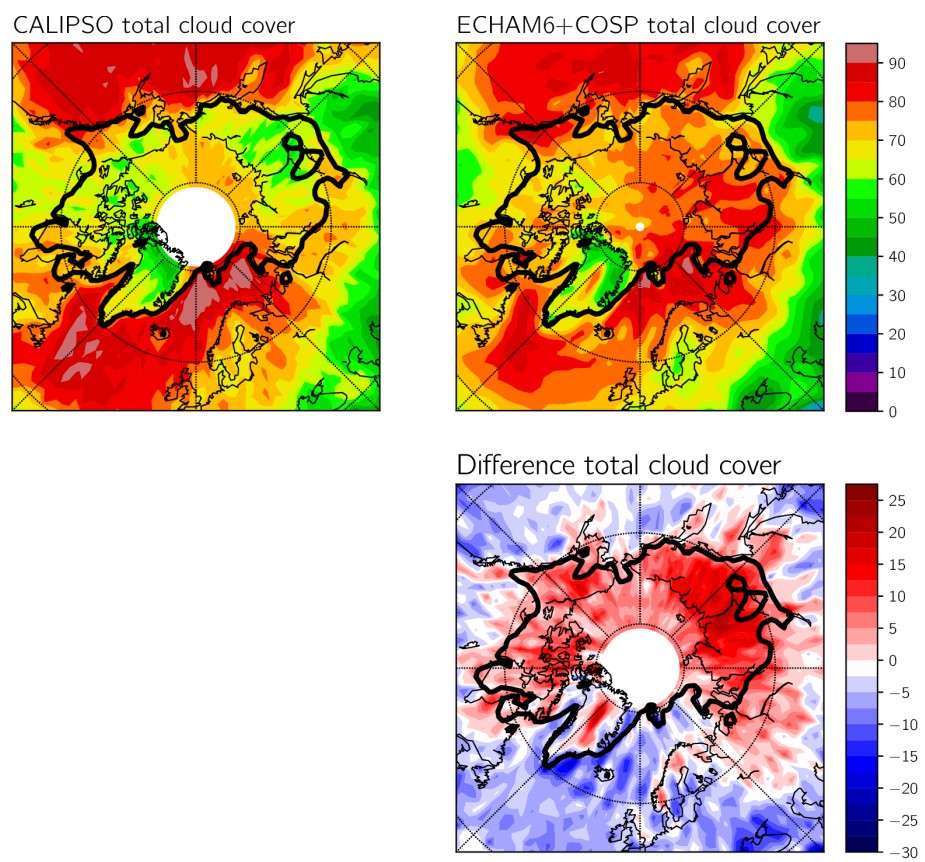

**Figure 2.** Top: Multi-year (2007-2011) mean total cloud cover as observed by CALIPSO and ECHAM6 + COSP. Bottom: Difference between the model and CALIPSO total cloud cover. Black line indicates regions with sea-ice cover greater then 50% or snow cover greater than 2 cm.

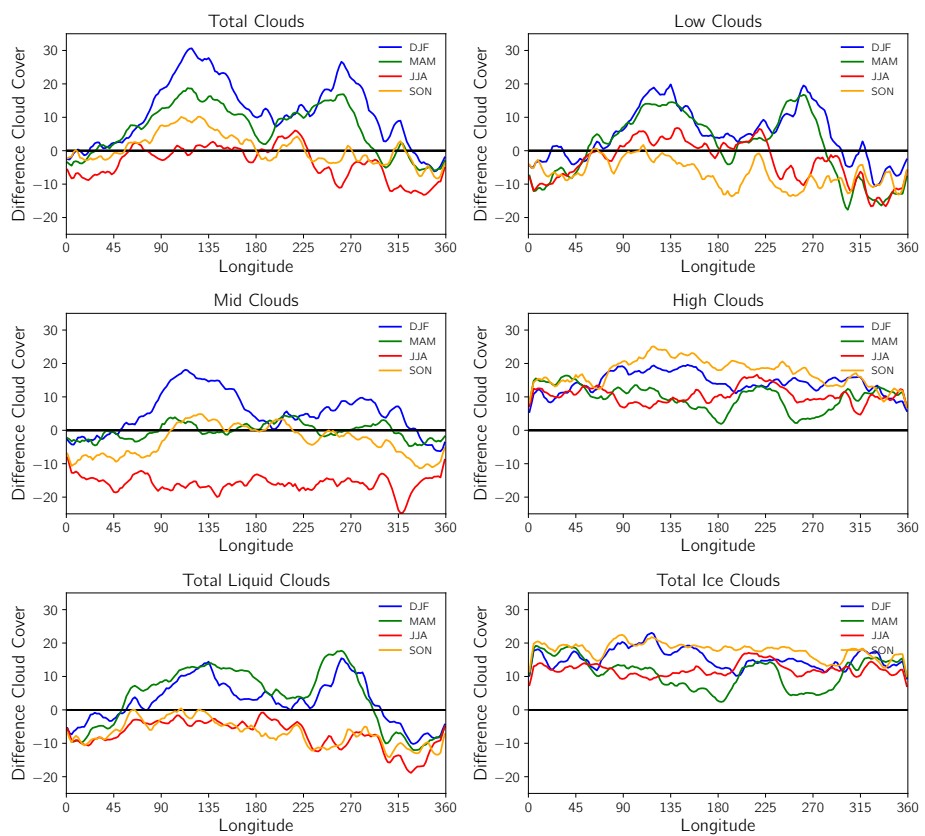

**Figure 3.** Meridional mean (60°N to 82°N) difference in cloud cover (model - satellite) between ECHAM6 + COSP and CALIPSO for total, low, mid and high clouds as well as difference in total liquid and total ice cloud cover.

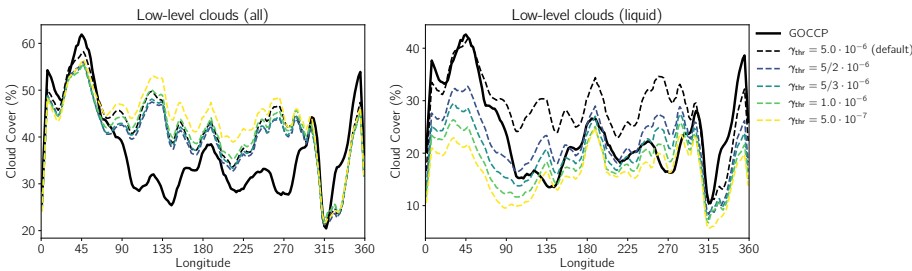

**Figure 4.** Meridional mean (60°N to 82°N) low-level (left) and low-level liquid cloud cover for different settings of $\gamma_{thr}$ (unit of $\gamma_{thr}$ is kg m$^{-3}$)

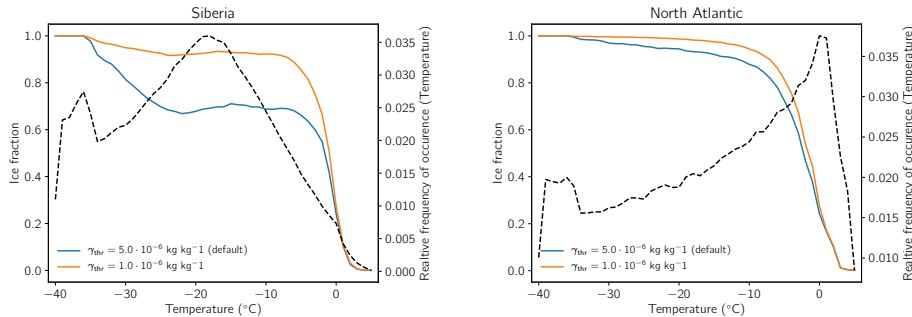

**Figure 5.** Temperature-binned, averaged ice fraction (IWC/(LWC+IWC)) in the North Atlantic (320-10°E / 50-70°N) and in Siberia (50-130°E / 50-70°N) . The dashed line shows the relative frequency of occurrence for the respective temperature bin.

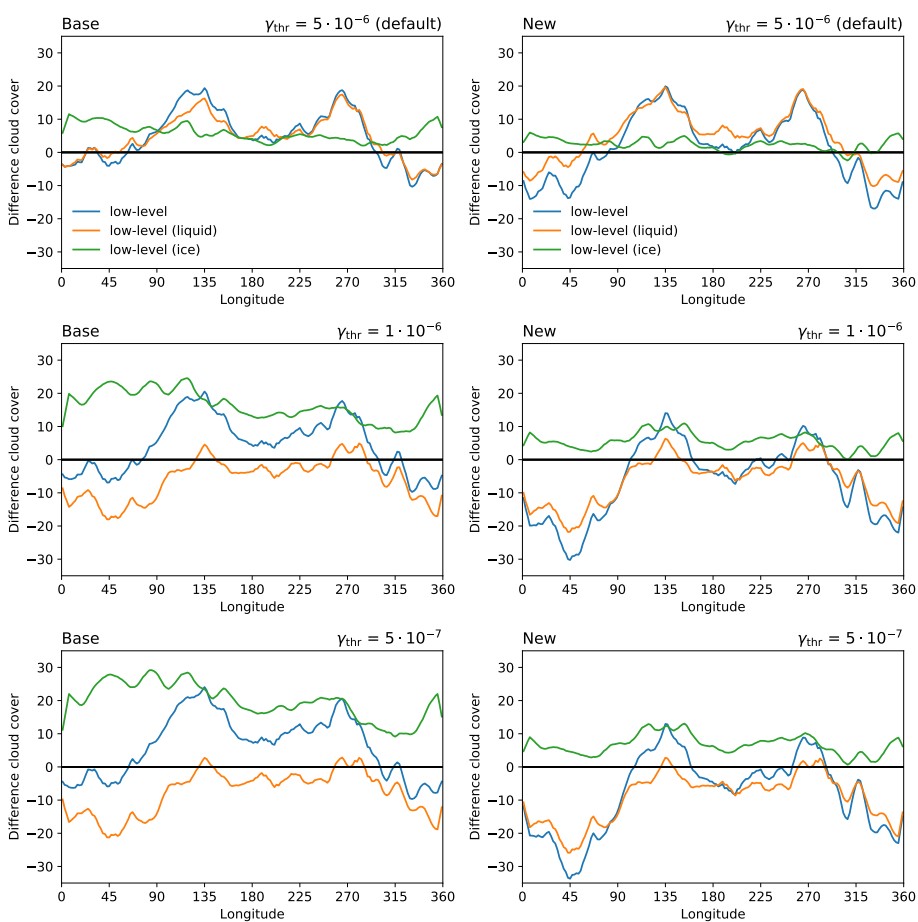

**Figure 6.** DJF low level cloud cover difference (all, liquid and ice clouds) to GOCCP for standard (Base) and modified (New) calculation of saturation water vapor pressure in the cloud cover scheme for different values of $\gamma_{\mathrm{thr}}$.

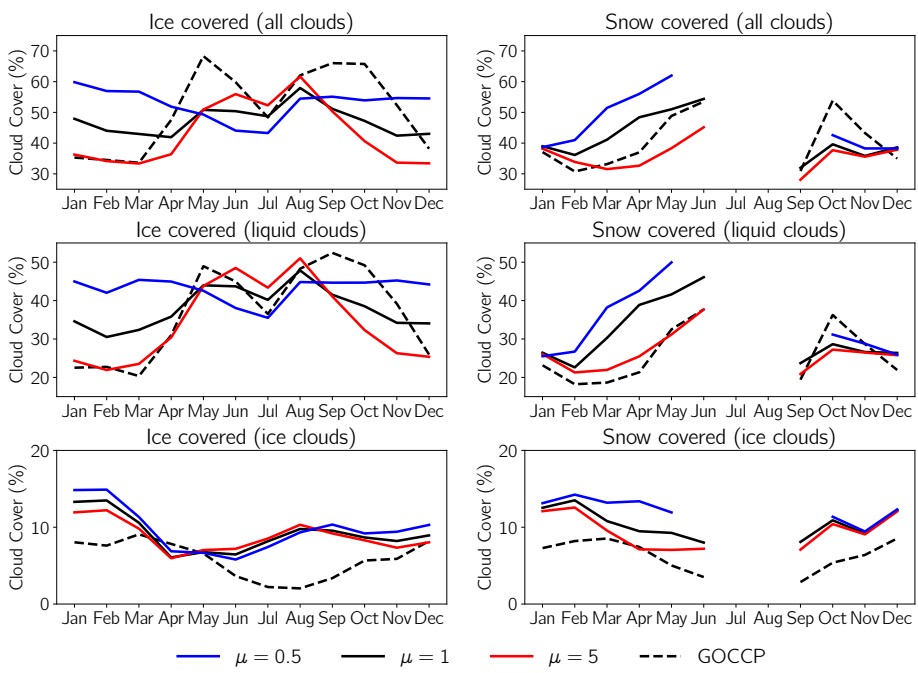

**Figure 7.** North hemispheric low-level cloud cover from ECHAM6+COSP over sea ice (left) and snow (right) covered surface for different strengths of surface mixing for all clouds (top), liquid clouds (middle) and ice clouds (bottom). The respective GOCCP cloud cover is shown for comparison.

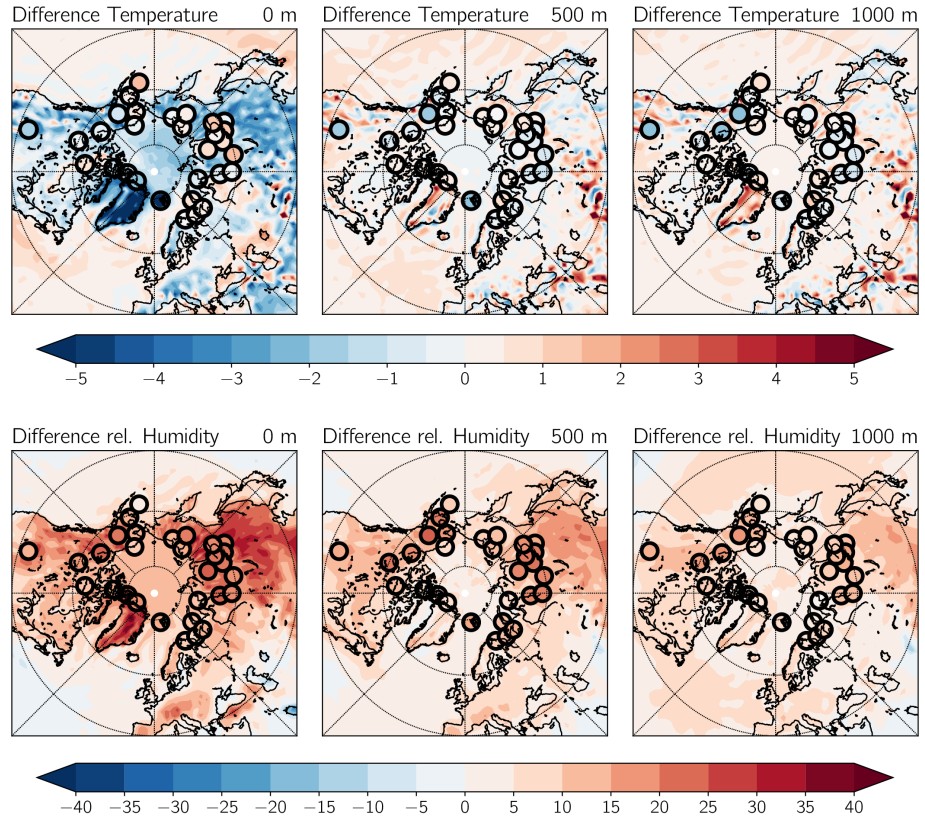

**Figure A1.** Vertical profiles of temperature and relative humidity differences between ECHAM6 and ERA-Interim averaged from 2007 to 2010. Filled circles show the same difference for profiles derives from radiosonde data. The vertical coordinate is height above ground level (AGL).