# Peer review of "Arctic clouds in ECHAM6 and their sensitivity to cloud microphysics and surface fluxes"

_Atmospheric Chemistry and Physics, 2018_

## Referee Comment (RC1) · Anonymous Referee #1 · 10 Dec 2018

The manuscript by Kretzschmar et al. shows a positive bias in cloud cover over the Arctic from global atmospheric model ECHAM6 with comparison with CALIPSO, and studies the possible causes of this difference, and presents their efforts to remove this bias with different parameterization in the model. The efforts include adjustment of moisture/heat exchange between surface and atmosphere, and tune of the effectiveness of Wegener-Bergeron-Findeisen process, and to allow supersaturation with respect to ice with a new parameterization of saturation water vapor pressure. The paper is generally well written and concise, which I particularly appreciate. The primary concern I see is the effectiveness of the new parameterization of the saturation water vapor pressure over both sea ice/snow and open water, which the authors may need a better presentation of their results. My recommendation is that the manuscript

needs be revised prior to publication to better present the effectiveness of their new approaches. More details follow. Major comments: 1. By allowing super saturation regarding to ice, the differences of low-level cloud between model and CALIPSO are somewhat reduced over sea ice/snow especially with smaller ice mixing ratio thresholds, as shown in Figure 6. The benefit is accompanied by the drawbacks that the differences over other areas become negative, with even more negative differences over open water, e.g. over the GIN seas and Barents Sea. As shown in Figure 1 and 2, such negative differences exist with original parameterization. So, the new parameterization may lead to another issue, underestimation of cloud cover, over the open water area, including the newly open water in the Arctic in summer and autumn. The authors mentioned the reduced condensation removal by precipitation may solve this, which may trigger other issues. The authors may want to clarify this in the manuscript. 2. The adjustment of surface/atmosphere heat/moisture strength seems working fine to me. In the manuscript, the authors said "For sea ice covered surfaces, ....while only minor changes in the cloud cover bias are found for summer". (Line 18- Line 23, page 7). I see the cloud cover after the adjustment agrees with CALIPSO really well over sea ice with scaling factor 5 as shown in Figure 4. The differences in winter are small, and the apparent difference in the later summer and autumn might be due to the CALIPSO shows more cloud cover over newly open water in the Arctic Ocean, while the model cloud cover are over sea ice only. This suggests this adjustment somehow works, even though the mixing is already too strong over the sea ice, as the authors discussed. 3. Model has positive bias in cloud cover, especially in low-level cloud, when compared to CALIPSO. CALIPSO has low cloud amount bias when compared to surface based observations, as studied by Blanchard et al. (2014) and Liu et al. (2017). The overestimation over sea ice may be appearing as significant considering CALIPSO's underestimation of low-level cloud. Specific comments: 1. Line 6 page 1, this overestimation is also due to overestimation of high-level cloud. 2. Line 21 page 2, please spell Acronym out at its first appearance, like CALIPSO; also after the first appearance, there is no need to spell it out again, like COSP. 3. Line 33-35 on

page 3, I am wondering what SST and ice concentration data you used in your model run? 4. Line 11-12 page 4, I am wondering how you are able to divide each model grid box into 40 subcolumns? 5. Line 19-20 page 4, you have model runs from 2007-2010, when sea ice extent in the late summer and autumn were significantly reduced. The cloud cover is greatly affected by this. It would be good to have the model runs from other years without such sea ice extent changes, which was not available in this study due to the computational cost as the authors pointed out. 6. Last line on page 4, consider changing "higher" to "greater". 7. In the 1st paragraph of section 3, you might want to point out there is underestimation of cloud cover over open water. 8. Line 11-12 on page 6, unless you show there is no humidity bias over other surface types, this claim may not be valid. 9. Line 18-23 on page 7, it would be interesting to see the impacts of the adjustment on liquid and ice cloud cover. 10. Line 9-10 on page 8, the differences also include bias in high-cloud. 11. Line 19 on page 8, "below" should be "above" 12. Line 28-29 on page 8, how about the changes in low-level cloud? 13. Line 28-31 on page 10, please reword this sentence. References included in the comment: Blanchard, Y., Pelon, J., Eloranta, E. W., Moran, K. P., Delanoë, J., and Sèze, G.: A synergistic analysis of cloud cover and vertical distribution from A-Train and ground-based sensors over the high Arctic station EUREKA from 2006 to 2010, J. Appl. Meteorol. Climatol., 53, 2553–2570, 2014. Liu, Y., Shupe, M. D., Wang, Z., and Mace, G.: Cloud vertical distribution from combined surface and space radar–lidar observations at two Arctic atmospheric observatories, Atmos. Chem. Phys., 17, 5973-5989, https://doi.org/10.5194/acp-17-5973-2017, 2017.

---

## Referee Comment (RC2) · Anonymous Referee #2 · 4 Jan 2019

This manuscript uses satellite observations from CALIPSO to evaluate Arctic cloud cover in ECHAM6. The authors found that low liquid cloud cover in the Arctic is biased high over surfaces covered by snow and ice in the default version of the model. They investigate two potential reasons for the high bias — the strength of surface heat fluxes and the impact of the Wegener-Bergeron-Findeisen (WBF) process. The authors conclude that surface heat fluxes are too strong in the default version of the model and that they can instead decrease their high bias in Arctic low liquid cloud cover by allowing for slight supersaturation with respect to ice in their cloud cover scheme, which in turn impacts the WBF process in ECHAM6.

I have numerous concerns about the manuscript that are primarily related to the methodology and conclusions drawn by the authors. My comments are below.

[Figure]

- The description of the observational dataset does not contain a discussion of observational uncertainties associated with CALIPSO/GOCCP. Namely, lidar beam attenuation is particularly problematic in the Arctic, where many clouds are optically thick, liquid, low-lying and precipitate snow. When compared to ground-based observations in the Arctic, CALIOP cannot "see" clouds in the lowest few kilometers (see e.g. Liu et al. (2017)) and the difference with GOCCP can be quite substantial especially over the Greenland ice sheet (Lacour et al. (2017)). This was also noted to be problematic in Cesana et al. (2012), and mostly affects precipitating ice underneath optically thick liquid clouds. I worry that the authors claim of a high bias in low, liquid clouds in the Arctic and their comparison for ice clouds may be inaccurate for the aforementioned reasons. The disadvantage of ground-based remote sensing observations, of course, is their lack of spatial coverage. I would still, however, recommend that the authors incorporate Arctic ground-based remote sensing observations from a few sites collocated with GOCCP to get an idea of potential biases that might impact their conclusion. Furthermore, the description of the observational dataset also does not mention the vertical resolution and criteria used for phase discrimination in the GOCCP product. Were daytime and nighttime data used? What timeframe was used? Were data before prior to the change in nadir-viewing angle used? How were oriented crystals handled?

- The authors note that ECHAM6 mixes too strongly in the Arctic and instead decide to turn to the model's parameterization of the WBF process instead to attempt to remedy the bias in Arctic cloud cover. To this end, the authors increased the efficiency of the WBF process by decreasing the threshold of in-cloud ice water mixing ratio required to activate the depositional growth of ice. However, it appears that the authors are unaware that ECHAM6 (Lohmann and Neubauer (2018)), like many other climate models (Komurcu et al. (2014), Cesana et al. (2015), McCoy et al. (2016)), underestimates the proportion of liquid to ice

in mixed-phase clouds. Decreasing the efficiency of the WBF process would only exacerbate this underestimate (Tan and Storelvmo (2016), Lohmann and Neubauer (2018)), which could also affect the climate sensitivity of the model (Tan et al. (2016), Lohmann and Neubauer (2018)). Thus, although the bias in cloud cover might be remedied, the partitioning of cloud phase would be further exacerbated. I would recommend the authors to look into how cloud thermodynamic phase is affected in the model before retuning the WBF process, which previous studies have already shown to be too efficient in climate models, including ECHAM6. Why do the authors choose to focus on the WBF process? Why not ice nucleation for example, which also plays an important role in Arctic radiation (Prenni et al. (2007), Xie et al. (2013))?

• The authors note that although there were improvements to Arctic low liquid cloud cover by increasing the efficiency of the WBF process, total cloud fraction remained overestimated. To this end, the authors then modified the cloud cover scheme to allow for slight supersaturation with respect to ice in the model (their "NEW" experiments). The authors seem to point out in the main text that cloud although some of the high bias in low-cloud fraction is reduced in their NEW simulations, new low-biases in low-cloud cover are introduced. Although improvements to the high bias in low-cloud fraction were highlighted in the abstract and conclusions, they authors fail to mention that there appears to be a simultaneous introduction of a new low bias in low-cloud cover. In fact, this low bias in Arctic low-cloud fraction was already shown for the CAM5 model (Kay et al. (2016)), which allows for supersaturation with respect to ice (Gettelman et al. (2010)). Therefore, the authors' parameterization does not seem to entirely solve the problem of the high bias in low-clouds in the Arctic, and the problem now reduces to an issue known to already exist in another model.

Also, although their temperature-weighted scheme for saturation vapour pressure may be new to the ECHAM6 model, it is not a new concept to climate models.

[Figure]

Please cite previous work that have used similar weighting schemes in the calculation of saturation vapour pressure:

1. Fowler, Laura D., David A. Randall, and Steven A. Rutledge. Liquid and ice cloud microphysics in the CSU general circulation model. Part 1: Model description and simulated microphysical processes. Journal of climate 9.3 (1996): 489-529.

2. Lord, Stephen J., Hugh E. Willoughby, and Jacqueline M. Piotrowicz. Role of a parameterized ice-phase microphysics in an axisymmetric, nonhydrostatic tropical cyclone model. Journal of the atmospheric sciences 41.19 (1984): 2836-2848.

3. Wood, Robert, and Paul R. Field. Relationships between total water, condensed water, and cloud fraction in stratiform clouds examined using aircraft data. Journal of the atmospheric sciences 57.12 (2000): 1888-1905.

- Section 3: It seems to me that there is a "chicken and egg" game when using observations of the vertical profiles of temperature and humidity to establish a cause for high bias in low liquid clouds in the model. Low-clouds can in turn affect temperature and relative humidity, so how can one establish the cause for the low-cloud bias?

References:

1. Liu, Yinghui, et al. Cloud vertical distribution from combined surface and space radar–lidar observations at two Arctic atmospheric observatories. Atmospheric Chemistry and Physics (Online) 17.9 (2017).

2. Lacour, Adrien, et al. Greenland clouds observed in CALIPSO-GOCCP: Comparison with ground-based Summit observations. Journal of Climate 30.15 (2017): 6065-6083.

3. Cesana, Gregory, et al. Ubiquitous low-level liquid-containing Arctic clouds: New observations and climate model constraints from CALIPSO-GOCCP. Geophysical Research Letters 39.20 (2012).

4. Lohmann, Ulrike, and David Neubauer. The importance of mixed-phase and ice clouds for climate sensitivity in the global aerosol–climate model ECHAM6-HAM2. Atmospheric Chemistry and Physics 18.12 (2018): 8807-8828.

5. Komurcu, Muge, et al. Intercomparison of the cloud water phase among global climate models. Journal of Geophysical Research: Atmospheres 119.6 (2014): 3372-3400.

6. Cesana, G., et al. Multimodel evaluation of cloud phase transition using satellite and reanalysis data. Journal of Geophysical Research: Atmospheres 120.15 (2015): 7871-7892.

7. McCoy, Daniel T., et al. On the relationships among cloud cover, mixed-phase partitioning, and planetary albedo in GCMs. Journal of Advances in Modeling Earth Systems 8.2 (2016): 650-668.

8. Tan, Ivy, and Trude Storelvmo. Sensitivity study on the influence of cloud microphysical parameters on mixed-phase cloud thermodynamic phase partitioning in CAM5. Journal of the Atmospheric Sciences 73.2 (2016): 709-728.

9. Tan, Ivy, Trude Storelvmo, and Mark D. Zelinka. Observational constraints on mixed-phase clouds imply higher climate sensitivity. Science 352.6282 (2016): 224-227.

10. Prenni, Anthony J., et al. Can ice-nucleating aerosols affect Arctic seasonal climate?. Bulletin of the American Meteorological Society 88.4 (2007): 541-550.

11. Xie, Shaocheng, et al. Sensitivity of CAM5-simulated Arctic clouds and radiation to ice nucleation parameterization. Journal of Climate 26.16 (2013): 5981-5999.

12. Kay, Jennifer E., et al. Evaluating and improving cloud phase in the Community Atmosphere Model version 5 using spaceborne lidar observations. Journal of Geophysical Research: Atmospheres 121.8 (2016): 4162-4176.

13. Gettelman, Andrew, et al. Global simulations of ice nucleation and ice supersaturation with an improved cloud scheme in the Community Atmosphere Model. Journal of Geophysical Research: Atmospheres 115.D18 (2010).

Minor Comments:

- Abstract, line 9: "Phase partitioning" typically refers to mass ratio or frequency ratio defined as liquid/(liquid + ice) in mixed-phase clouds within a grid cell or specified domain. Here, the authors refer to the ratio of total low liquid cloud cover to total cloud cover. I recommend changing the terminology to avoid confusion.

- I suggest changing the title of Section 2.1 to "GOCCP" to reflect the fact that this CALIPSO-derived product was used in the analysis.

- Page 2, lines 20-23: I would also mention the advantage that active satellites are also able to provide vertical profiles of clouds.

- Page 5, lines 10-13: If the mid-level cloud bias is similar to the low-cloud bias because of how low- and mid-level clouds are defined, then shouldn't that mean that the bias in mid-level clouds for JJA should resemble the bias for high clouds? It does not appear to.

- Page 5, line 20: This is an overstatement without formal proof. I would suggest replace "is" with "appears to be".

- Page 6, lines 19-22: This is an interesting hypothesis that may or may not be true. I would be more careful in emphasizing that the statement is speculative.

- Page 8, line 13: Please add a reference for the WBF process and note the ways in which models simplify it (e.g. lack of dependence of vertical velocity). Please see Korolev (2007).

- Page 8, line 21: "will" should go in front of "depositional".

- Page 10, Lines 11-12: Please specify that this the overestimate is with respect to GOCCP.

- Page 11, lines 15-17: I disagree with this statement. The Karcher and Lohmann paper refers to cirrus clouds. In mixed-phase clouds, where liquid and ice clouds coexist and the WBF process occurs, the cloud may not necessarily glaciate immediately and will instead depend on how the liquid and ice are spatially distributed within the cloud (Tan and Storelvmo (2016)).

- Page 12, Line 17: "reduce to" ⇒ "reduce the"

- Page 12, line 18: Please specify that supersaturation is with respect to ice.

- Figure 4: strength ⇒ strength

- Figure 5: Please consider labelling the first value as the "default" value of the model in the legend of this figure for easy reference.

- Please remove all instances of "the" in front of "Arctic amplification".

Reference:

1. Korolev, Alexei. "Limitations of the Wegener–Bergeron–Findeisen mechanism in the evolution of mixed-phase clouds." Journal of the Atmospheric Sciences 64.9 (2007): 3372-3375.

---

## Author Comment (AC1) · 29 Mar 2019

**Response to Referee #1**

*The manuscript by Kretzschmar et al. shows a positive bias in cloud cover over the Arctic from global atmospheric model ECHAM6 with comparison with CALIPSO, and studies the possible causes of this difference, and presents their efforts to remove this bias with different parameterization in the model. The efforts include adjustment of moisture/heat exchange between surface and atmosphere, and tune of the effectiveness of Wegener-Bergeron-Findeisen process, and to allow supersaturation with respect to ice with a new parameterization of saturation water vapor pressure. The paper is generally well written and concise, which I particularly appreciate. The primary concern I see is the effectiveness of the new parameterization of the saturation water vapor pressure over both sea ice/snow and open water, which the authors may need a better presentation of their results. My recommendation is that the manuscript needs be revised prior to publication to better present the effectiveness of their new approaches.*

We thank the reviewer for the constructive comments that helped to improve the manuscript.

**Major comments**

*1. By allowing super saturation regarding to ice, the differences of low-level cloud between model and CALIPSO are somewhat reduced over sea ice/snow especially with smaller ice mixing ratio thresholds, as shown in Figure 6. The benefit is accompanied by the drawbacks that the differences over other areas become negative, with even more negative differences over open water, e.g. over the GIN seas and Barents Sea. As shown in Figure 1 and 2, such negative differences exist with original parameterization. So, the new parameterization may lead to another issue, underestimation of cloud cover, over the open water area, including the newly open water in the Arctic in summer and autumn. The authors mentioned the reduced condensation removal by precipitation may solve this, which may trigger other issues. The authors may want to clarify this in the manuscript.*

The idea that reduced condensation removal by precipitation may solve this issue was purely speculative and we did not conduct sensitivity studies to this end and therefore we removed this statement from the manuscript. In the revised version of the manuscript, we try to more clearly point out why a temperature-weighted scheme for saturation vapor pressure in combination with an increased efficiency of the WBF process introduces a negative bias in low clouds. As the amount of low-level ice clouds remains more or less constant for different values of $\gamma_{\mathrm{thr}}$, the amount of liquid clouds strongly decreases and therefore also the amount of clouds in general. The decrease in liquid clouds is mainly caused by the more efficient WBF processes which more efficiently turns liquid into ice clouds over continents compared to oceanic regions. In the standard setup of ECHAM, liquid clouds are already biased low in those regions which is even further enhanced by a more effective WBF process. As liquid clouds seem to react rather sensitively to a more effective WBF process, even minor changes of $\gamma_{\mathrm{thr}}$ can have strong effects on the amount of liquid clouds and we think that setting $\gamma_{\mathrm{thr}}$ to $2.5 \cdot 10^{-6}$ kg m$^{-3}$ might already be a reasonable value to improve the WBF process. This value might be a good compromise between improving cloud cover over snow and ice covered surfaces by simultaneously not further worsen clouds in other regions. These new explanations are now in the manuscript in order to respond to the reviewer's remark.

*2. The adjustment of surface/atmosphere heat/moisture strength seems working fine to me. In the manuscript, the authors said "For sea ice covered surfaces, ...while only minor changes in the cloud cover bias are found for summer". (Line 18- Line 23, page 7). I see the cloud cover after the adjustment agrees with CALIPSO really well over sea ice with scaling factor 5 as shown in Figure 4. The differences in winter are small, and the apparent difference in the later summer and autumn might be due to the CALIPSO shows more cloud cover over newly open water in the Arctic Ocean, while the model cloud cover are over sea ice only. This suggests this adjustment somehow works, even though the mixing is already too strong over the sea ice, as the authors discussed.*

We agree with the reviewer that increased mixing seems indeed be a good way of tuning Arctic clouds. As requested by the reviewer also in one of the minor comments below, we added a new, more detailed discussion of the effect of this adjustment on liquid and ice clouds. Increased mixing was also able to improve cloud phase as the liquid bias in winter is now also reduced, which further shows that this might be a good option to improve clouds. In the revised version of the manuscript, we try to emphasize the positive effect of increased mixing on cloud cover, even though we still think that it might be questionable whether one can physically justify such a measure as the model already mixes too strongly in the Arctic with its stable boundary layers in comparison to surface observations.

*3. Model has positive bias in cloud cover, especially in low-level cloud, when compared to CALIPSO. CALIPSO has low cloud amount bias when compared to surface based observations, as studied by Blanchard et al. (2014) and Liu et al. (2017). The overestimation over sea ice may be appearing as significant considering CALIPSO's underestimation of low-level cloud.*

We revised the description of the CALIPSO-COCCP dataset. Section 2 now contains a more detailed description of the observational dataset (i.e. cloud detection thresholds, information on vertical resolution, phase discrimination). In the revised version of the manuscript, a more detailed review of uncertainties and issues for retrieving clouds in the Arctic using CALIPSO-GOCCP is included (i.e. lidar attenuation by liquid clouds, cloud detection thresholds that might not be representative for Arctic region and also possible effects of spatio-temporal sampling of satellite data). Nevertheless, we think that our claim of an overestimated low-level cloud fraction in ECHAM6 is valid. We compared modeled (ECHAM+COSP minus ECHAM) to observed (GOCCP minus ground based observations) cloud cover profile differences and see a similar underestimation for modeled clouds when using a satellite simulator compared to the cloud fraction from ECHAM6's cloud cover scheme. Even though comparing modeled and observed difference in cloud cover profiles is not an "apples-to-apples" comparison (because of different definitions of what is a cloud), we see that COSP derived cloud properties mimic real world issues of the actual lidar. Therefore, the reported overestimation of low-level clouds in the model is a "real" signal and not just due the observational issues in the GOCCP dataset.

**Specific comments**

*1. Line 6 page 1, this overestimation is also due to overestimation of high-level cloud.*

We added the explanation that the overestimation of total cloud cover is due to an overestimation of low- and high-level clouds to the abstract.

*2. Line 21 page 2, please spell Acronym out at its first appearance, like CALIPSO; also after the first appearance, there is no need to spell it out again, like COSP.*

Done.

*3. Line 33-35 on page 3, I am wondering what SST and ice concentration data you used in your model run?*

We use monthly observations of sea surface temperature and sea ice concentration from the AMIP II dataset. We added this to the manuscript.

*4. Line 11-12 page 4, I am wondering how you are able to divide each model grid box into 40 subcolumns?*

In the revised version of the manuscript we elaborate more on how those subcolums are created.

*5. Line 19-20 page 4, you have model runs from 2007-2010, when sea ice extent in the late summer and autumn were significantly reduced. The cloud cover is greatly affected by this. It would be*

*good to have the model runs from other years without such sea ice extent changes, which was not available in this study due to the computational cost as the authors pointed out.*

The reviewer is right that this introduces a complication. But since the observations are for the same period, and since the biases are widespread, we think that the conclusions are valid. In any case, since CALIPSO and CloudSat are available only since 2006, there is no possibility to go for another period for the evaluation.

*6. Last line on page 4, consider changing "higher" to "greater".*

Done

*7. In the 1st paragraph of section 3, you might want to point out there is underestimation of cloud cover over open water.*

Done.

*8. Line 11-12 on page 6, unless you show there is no humidity bias over other surface types, this claim may not be valid.*

We additionally show from ERA-Interim to also have information on temperature and humidity profiles on a wider spatial scale to show that there is a difference between snow/ice covered regions and not snow/ice covered regions. Looking at relative humidity, ECHAM6 seems to generally overestimate it over the continents, but this overestimation is most strongly pronounced in those regions we observed the strongest positive biases in low-level clouds, which make us confident that this overestimation actually exists.

*9. Line 18-23 on page 7, it would be interesting to see the impacts of the adjustment on liquid and ice cloud cover.*

In the revised version of the manuscript, we added the impacts of the adjustment on liquid and ice cloud cover. As we already stated above, the approach of increased mixing seems promising as this measure not only reduces the cloud cover bias of low-level clouds but also addresses helps to reduce the overestimated bias of liquid clouds.

*10. Line 9-10 on page 8, the differences also include bias in high-cloud.*

The revised manuscript now explicitly points the reader to this fact.

*11. Line 19 on page 8, "below" should be "above"*

Using "below" in this sentence is correct, as we refer to condensation. Nevertheless, we see that this sentence can be misunderstood and modified it to be better understandable.

*12. Line 28-29 on page 8, how about the changes in low-level clouds?*

With total cloud cover, we mean total, low-level cloud cover. To avoid confusion, we now just call it "low cloud cover".

*13. Line 28-31 on page 10, please reword this sentence.*

Done.

---

## Author Comment (AC2) · 29 Mar 2019

**Response to Referee #2**

*This manuscript uses satellite observations from CALIPSO to evaluate Arctic cloud cover in ECHAM6. The authors found that low liquid cloud cover in the Arctic is biased high over surfaces covered by snow and ice in the default version of the model. They investigate two potential reasons for the high bias the strength of surface heat fluxes and the impact of the Wegener-Bergeron-Findeisen (WBF) process. The authors conclude that surface heat fluxes are too strong in the default version of the model and that they can instead decrease their high bias in Arctic low liquid cloud cover by allowing for slight supersaturation with respect to ice in their cloud cover scheme, which in turn impacts the WBF process in ECHAM6. I have numerous concerns about the manuscript that are primarily related to the methodology and conclusions drawn by the authors. My comments are below.*

We thank the reviewer for the constructive comments.

**Major comments**

*The description of the observational dataset does not contain a discussion of observational uncertainties associated with CALIPSO/GOCCP. Namely, lidar beam attenuation is particularly problematic in the Arctic, where many clouds are optically thick, liquid, low-lying and precipitate snow. When compared to ground-based observations in the Arctic, CALIOP cannot see clouds in the lowest few kilometers (see e.g. Liu et al. (2017)) and the difference with GOCCP can be quite substantial especially over the Greenland ice sheet (Lacour et al. (2017)). This was also noted to be problematic in Cesana et al. (2012), and mostly affects precipitating ice underneath optically thick liquid clouds. I worry that the authors claim of a high bias in low, liquid clouds in the Arctic and their comparison for ice clouds may be inaccurate for the aforementioned reasons. The disadvantage of ground-based remote sensing observations, of course, is their lack of spatial coverage. I would still, however, recommend that the authors incorporate Arctic ground-based remote sensing observations from a few sites collocated with GOCCP to get an idea of potential biases that might impact their conclusion.*

In the revised version of the manuscript, a more detailed review of the uncertainties related to the GOCCP dataset is included (i.e. lidar attenuation by liquid clouds, cloud detection thresholds that might not be representative for Arctic region and also possible affects of spatio-temporal sampling of satellite data).
Nevertheless, we think that our conclusion of an overestimated low-level cloud fraction in ECHAM6 is still valid. The GOCCP dataset is based on satellite retrievals and is not directly comparable to ground observations or to model output. In order to make our model results comparable to the GOCCP dataset we use the COSP satellite simulator. In the revised manuscript, we compare modeled (ECHAM6+COSP minus ECHAM6) to observed (GOCCP minus ground based observations) cloud cover profile differences and see a similar underestimation for modeled clouds when using a satellite simulator (ECHAM6+COSP) compared to the cloud fraction form ECHAM6's cloud cover scheme. While comparing modeled and observed differences in cloud cover profiles is not an "apples-to-apples" comparison (because of different definitions of what is a cloud), this demonstrates that COSP derived cloud properties can mimic real world issues of the spaceborne lidar. Therefore, the reported overestimation of low-level clouds in the model is a "real" signal and not just due the observational issues in the GOCCP dataset.

*Furthermore, the description of the observational dataset also does not mention the vertical resolution and criteria used for phase discrimination in the GOCCP product. Were daytime and nighttime data used? What timeframe was used? Were data before prior to the change in nadir-viewing angle used? How were oriented crystals handled?*

In light of this remark by the reviewer, we revised the description of the CALIPSO-COCCP dataset. Section 2 now contains a more detailed description of the observational dataset (i.e. cloud detection thresholds, information on vertical resolution, phase discrimination). In Section 3, we now also state

that we use monthly averaged data for the same timeframe as the model simulations using both, day- and nightime overpasses. Concerning the change of the nadir pointing angle at the end of 2007, the period we used for evaluation of ECHAM6 (2007-2010) could be affected by that. This would mainly affect the retrieval of the cloud phase due to an effect on the depolarization ratios by horizontally oriented crystals. As COSP does not use any information on the shape of ice crystals from the model (as most models do not have information on the shape of the ice crystals), the effect of horizontally oriented crystals can be ignored at least from the model side.

*The authors note that ECHAM6 mixes too strongly in the Arctic and instead decide to turn to the models parameterization of the WBF process instead to attempt to remedy the bias in Arctic cloud cover. To this end, the authors increased the efficiency of the WBF process by decreasing the thresh- old of in-cloud ice water mixing ratio required to activate the depositional growth of ice. However, it appears that the authors are unaware that ECHAM6 (Lohmann and Neubauer(2018)), like many other climate models (Komurcu et al. (2014), Cesana et al.(2015), McCoy et al. (2016)), underes- timates the proportion of liquid to ice in mixed-phase clouds. Decreasing the efficiency of the WBF process would only exacerbate this underestimate (Tan and Storelvmo (2016), Lohmann and Neubauer (2018)), which could also affect the climate sensitivity of the model (Tan et al. (2016), Lohmann and Neubauer (2018)).*

Citing Lohmann and Neubauer (2018), the reviewer states that ECHAM6, like many other climate models, underestimates the proportion of liquid to ice in mixed-phase clouds. We would like to point out that Lohmann and Neubauer (2018) did not use the ECHAM6 Stevens et al. (2013), but used ECHAM6-HAM2 Zhang et al. (2012). Even though both models share a lot of their physical param- eterizations, they significantly differ in the microphysical parametrizations. While ECHAM6 employs a single-moment scheme, ECHAM6-HAM2 uses a more sophisticated double-moment scheme. Even though both microphysical schemes stem from a common predecessor, they considerably vary in a lot of microphysical processes. One has therefore be careful when comparing ECHAM6-HAM2 to ECHAM6. Figure 3 in Lohmann and Neubauer (2018) shows the fraction of supercooled liquid clouds for ECHAM6-HAM2 (as well as for a number of sensitvity studies) on a global average. While on global average ECHAM6-HAM2 might underestimate this fraction, this figure does not show the fraction of super- cooled liquid clouds in the Arctic. Komurcu et al. (2014) provides zonal-mean averages of supercooled liquid cloud fraction for different cloud top temperatures for ECHAM6-HAM2 (see their Figure 4) and for temperatures at or below -30° C, ECHAM6-HAM2 overestimates the amount of supercooled liquid clouds for high latitudes, even though by not much.
Figure 5 in Cesana et al. (2015) provides a similar zonal-mean, temperature binned supercooled liquid cloud fraction for MPI-ESM Giorgetta et al. (2013), which is the coupled version of ECHAM6, and a similar overestimation of supercooled liquid shows for MPI-ESM in the Arctic (compared to GOCCP at temperatures below -30° C). This overestimation of liquid cloud fraction in the lower part of the mixed-phase temperature regime is consistent with the fact that the overestimation of liquid cloud is only simulated in winter (DJF) and spring (MAM) where such cold temperatures can occur in high latitudes. Additionally, while being positively biased in high latitudes, MPI-ESM slightly underesti- mates the amount of supercooled liquid in the clouds in the mid-latitudes and in the tropics (see their Figure 6) even though not by much.

*Thus, although the bias in cloud cover might be remedied, the partitioning of cloud phase would be further exacerbated. I would recommend the authors to look into how cloud thermodynamic phase is affected in the model before retuning the WBF process, which previous studies have already shown to be too efficient in climate models, including ECHAM6.*

The reviewer is correct that even though the bias in liquid cloud fraction might be remedied by a stronger WBF processes, the effects of this measure on the actual (mass) phase partitioning (IWC/(LWC+IWC)) might be different. To this end, we follow the reviewer's advice and look into how cloud thermodynam- ical phase is affected before retuning the model. There is no observational product that can provide

both, liquid and ice water content, on a large enough scale to compare it to a GCM. This is also the reason why all the studies cited by the reviewer are trying to mimic frequency ratio fraction of the cloud phase that can be provided by CALIOP. A possible approach to evaluate cloud phase would be to look at liquid/ice water path which can be derived from MODIS. As stated in the introduction, using passive spaceborne sensors might be problematic due to the environmental conditions and also due to fact the Arctic clouds are often mixed-phase clouds, which further complicates the retrieval of cloud microphysical properties (Khanal and Wang, 2018). To obtain at least a rough estimate of how the ice (mass) fraction is affected by a stronger by a stronger WBF process in ECHAM6, we added a plot of temperature-binned average ice fraction over the North Atlantic and over Siberia (Figure 6 in the revised manuscript). For the ice fraction in Siberia, we find quite low ice fraction ($\sim 70\%$) in the temperature range between -25° C and -10° C. Comparing this to in-situ observation of ice fraction as provided by Korolev et al. (2017) such a "plateau" is not visible. Figure 5-14 in Korolev et al. (2017) shows a more gradual increase in ice fraction (decrease in liquid fraction) with decreasing temperature (which can be seen in the bins for high/low ice fraction) and we think that the more or less constant ice fraction in the model over Siberia is another indication of an overestimated amount of liquid clouds over snow/ice covered surface as has been stated in the manuscript. As the ice fractions from in-situ observations and the ice fractions from the model are on a completely different spatial scale, one nevertheless has to be careful when doing such a comparison. As we have shown in our conclusion, the TOA shortwave CRE seems to be biased low in MPI-ESM which might be another hint that there is more liquid water in the clouds, which would make them less reflective, so we think that a slightly stronger efficiency of the WBF and therefore an higher ice (mass) fraction can be justified.

*Why do the authors choose to focus on the WBF process? Why not ice nucleation for example, which also plays an important role in Arctic radiation (Prenni et al. (2007), Xie et al. (2013))?*

The reason why we focused on the WBF is twofold. Firstly, it has to be a process that is able to efficiently reduce the amount of cloud liquid water. We conducted a number of sensitivity studies and modified the strength of all processes that can affect the liquid water content and we found the WBF to be by far the most efficient one. It also can be seen from table 4 and 5 in Klaus et al. 2012 that only the WBF process ($\gamma_{thr}$) and the collection of cloud droplets by snow ($\gamma_4$) are able to do so. Not included in this table is heterogeneous freezing of cloud droplets, but we found that increasing its efficiency did not lead to strong enough reduction in liquid cloud cover over snow and ice covered surfaces. Secondly, what makes it appealing to tune this process is the fact that it is strongly simplified in ECHAM6. Due to efficiency in tuning the amout of ice in clouds, modifying the strength of this process is also often used to tune the model to bring it into radiative balance. This can be seen from the fact that this parameter can vary up to an order of magnitude for different horizontal resolutions in ECHAM6. These considerations are now explained in more detail in the revised manuscript.

*The authors note that although there were improvements to Arctic low liquid cloud cover by increasing the efficiency of the WBF process, total cloud fraction remained overestimated. To this end, the authors then modified the cloud cover scheme to allow for slight supersaturation with respect to ice in the model (their NEW experiments). The authors seem to point out in the main text that cloud although some of the high bias in low-cloud fraction is reduced in their NEW simulations, new low-biases in low-cloud cover are introduced. Although improvements to the high bias in low-cloud fraction were highlighted in the abstract and conclusions, they authors fail to mention that there appears to be a simultaneous introduction of a new low bias in low-cloud cover. In fact, this low bias in Arctic low-cloud fraction was already shown for the CAM5 model (Kay et al. (2016)), which allows for supersaturation with respect to ice (Gettelman et al. (2010)). Therefore, the author's parameterization does not seem to entirely solve the problem of the high bias in low-clouds in the Arctic, and the problem now reduces to an issue known to already exist in another model..*

In the revised version of the manuscript, we try to more clearly point out why a temperature-weighted scheme for saturation vapor pressure in combination with an increased efficiency of the WBF process

introduces an negative bias in low clouds. As the amount of low-level ice clouds remains more or less constant for different values of $\gamma_{thr}$, the amount of liquid clouds strongly decreases and therefore also the amount of clouds in general. The decrease in liquid clouds is mainly caused by the more efficient WBF processes which more efficiently turns liquid into ice clouds over continents compared to oceanic regions, it also affects clouds there. In the standard setup of ECHAM, liquid clouds are already biased low in those regions which is even further enhanced by a more effective WBF process. As liquid clouds seem to react rather sensitively to a more effective WBF process, only minor changes of $\gamma_{thr}$ can have strong effects on the amount of liquid clouds and we think that setting $\gamma_{thr}$ to $2.5 \cdot 10^{-6}$ kg m$^{-3}$ is already the best choice to improve WBF process. This value is the best compromise between improving cloud cover over snow and ice covered surfaces by simultaneously not further worsen clouds in other regions.

*Also, although their temperature-weighted scheme for saturation vapor pressure may be new to the ECHAM6 model, it is not a new concept to climate models. Please cite previous work that have used similar weighting schemes in the calculation of saturation vapor pressure.*

In the revised version of the manuscript, we now cite previous work that have used similar weighting schemes in the calculation of saturation vapour pressure.

*Section 3: It seems to me that there is a chicken and egg game when using observations of the vertical profiles of temperature and humidity to establish a cause for high bias in low liquid clouds in the model. Low-clouds can in turn affect temperature and relative humidity, so how can one establish the cause for the low-cloud bias?*

The reviewer is correct that no causal relationship can be established between a positive bias in low-level temperature and humidity and a positive cloud cover bias. Nevertheless, we believe that such biases in temperature and humidity can be an indicator of an overestimated cloud cover due to this two-way relationship that has been stated by the reviewer. We mainly used this comparison of vertical profiles to show that the reported cloud cover bias is not just due to possible uncertainties in GOCCP but is a real model problem. On request by the other reviewer, we additionally show data from ERA-Interim to also have information on temperature and humidity profiles on a wider spatial scale to show that there is a difference between snow/ice covered regions and water/open land. Looking at relative humidity, ECHAM6 seems to generally overestimate it over the continents, but this overestimation is most strongly pronounced in those regions we observed the strongest positive biases in low-level clouds, which make us confident that this overestimation actually exists.

**Minor comments**

*Abstract, line 9: Phase partitioning" typically refers to mass ratio or frequency ratio defined as liquid/(liquid + ice) in mixed-phase clouds within a grid cell or specified domain. Here, the authors refer to the ratio of total low liquid cloud cover to total cloud cover. I recommend changing the terminology to avoid confusion.*

We replaced "Improvements in the phase partitioning of Arctic low-level clouds" with "Improvements on the overestimated Arctic low-level liquid cloud cover"

*I suggest changing the title of Section 2.1 to GOCCP" to reflect the fact that this CALIPSO-derived product was used in the analysis.*

In the revised manuscript, we replaced all instances of CALIPSO with GOCCP and completely revised section describing GOCCP.

*Page 2, lines 20-23: I would also mention the advantage that active satellites are also able to provide vertical profiles of clouds.*

We mentioned that actives satellites can provide vertical profiles of clouds which cannot be provided by passive satellites.

*Page 5, lines 10-13: If the mid-level cloud bias is similar to the low-cloud bias because of how low- and mid-level clouds are defined, then shouldnt that mean that the bias in mid-level clouds for JJA should resemble the bias for high clouds? It does not appear to.*

We misinterpreted the similarity of the mid-level cloud bias to the low-cloud bias and our explanation does not hold. We therefore looked into the vertical profile of clouds and at the altitude of the threshold for low-, mid- and high-clouds (see attached figure). The thresholds themselves vary only a little between summer and winter. The actual cause for the seasonal variation of the mid-cloud bias can be attributed to the vertical position of the generally overestimated high-clouds in ECHAM6. The vertical extent of the troposphere is influenced by the atmospheric temperature which cause the cirrus clouds to be present at lower altitudes in winter. The similarity to low-cloud stems form the fact the temperatures are colder over snow and ice covered surfaces, which cause the cirrus clouds to be simulated at even lower altitudes and therefore contributed more the mid-level clouds compared to oceanic regions. We correct our false claim in the revised manuscript.

*Page 5, line 20: This is an overstatement without formal proof. I would suggest replace is with appears to be.*

Done.

*Page 6, lines 19-22: This is an interesting hypothesis that may or may not be true. I would be more careful in emphasizing that the statement is speculative.*

We try to more clearly formulate that this statement is speculative in the revised manuscript.

*Page 8, line 13: Please add a reference for the WBF process and note the ways in which models simplify it (e.g. lack of dependence of vertical velocity). Please see Korolev (2007).*

We added a reference for the WBF process at its first mentioning at the end of section 3. We also stated how ECHAM6 simplifies the WBF process due to its lack of dependence of vertical velocity.

*Page 8, line 21: will" should go in front of depositional".*

Done.

*Page 10, Lines 11-12: Please specify that this the overestimate is with respect to GOCCP.*

We now specify that the overestimation is with respect to GOCCP.

*Page 11, lines 15-17: I disagree with this statement. The Karcher and Lohmann paper refers to cirrus clouds. In mixed-phase clouds, where liquid and ice clouds coexist and the WBF process occurs, the cloud may not necessarily glaciate immediately and will instead depend on how the liquid and ice are spatially distributed within the cloud (Tan and Storelvmo (2016)).*

We removed the reference to the Karcher and Lohmann form our manuscript as it indeed refers more to cirrus clouds. Nevertheless, the way that mixed-phase clouds are parameterized in ECHAM6 will eventually cause any liquid water to be depleted quite quickly, as the condensation is the only process the can produce water in the mixed-phase temperature regime. As soon as there is enough cloud ice present and it exceeds $\gamma_{thr}$, condensation does not take place any more and any liquid water will quite quickly either freeze or evaporated. This can indeed be considered not physical as the presently used implementation of condensation/deposition does not allow for simultaneous growth of liquid and ice within a cloud. ECHAM6 also has no information on the subgrid distribution of liquid and ice within a cloud which might prevent this rather rapid depletion of liquid water.

*Page 12, Line 17: reduce to" "reduce the"*

Done.

*Page 12, line 18: Please specify that supersaturation is with respect to ice.*

We now specify that supersaturation is with respect to ice.

*Figure 4: strength to strength*

Done.

*Figure 5: Please consider labelling the first value as the default value of the model in the legend of this figure for easy reference*

Done.

*Please remove all instances of the" in front of Arctic amplification".*

*Done.*

**References**

Cesana, G., Waliser, D. E., Jiang, X., and Li, J.-L. F. (2015). Multimodel evaluation of cloud phase transition using satellite and reanalysis data. *Journal of Geophysical Research: Atmospheres*, 120(15):7871–7892, doi:10.1002/2014JD022932.

Giorgetta, M. A., Jungclaus, J., Reick, C. H., Legutke, S., Bader, J., Bttinger, M., Brovkin, V., Crueger, T., Esch, M., Fieg, K., Glushak, K., Gayler, V., Haak, H., Hollweg, H.-D., Ilyina, T., Kinne, S., Kornblueh, L., Matei, D., Mauritsen, T., Mikolajewicz, U., Mueller, W., Notz, D., Pithan, F., Raddatz, T., Rast, S., Redler, R., Roeckner, E., Schmidt, H., Schnur, R., Segschneider, J., Six, K. D., Stockhause, M., Timmreck, C., Wegner, J., Widmann, H., Wieners, K.-H., Claussen, M., Marotzke, J., and Stevens, B. (2013). Climate and carbon cycle changes from 1850 to 2100 in mpi-esm simulations for the coupled model intercomparison project phase 5. *Journal of Advances in Modeling Earth Systems*, 5(3):572–597, doi:10.1002/jame.20038.

Khanal, S. and Wang, Z. (2018). Uncertainties in modis-based cloud liquid water path retrievals at high latitudes due to mixed-phase clouds and cloud top height inhomogeneity. *Journal of Geophysical Research: Atmospheres*, 123(19):11,154–11,172, doi:10.1029/2018JD028558.

Komurcu, M., Storelvmo, T., Tan, I., Lohmann, U., Yun, Y., Penner, J. E., Wang, Y., Liu, X., and Takemura, T. (2014). Intercomparison of the cloud water phase among global climate models. *Journal of Geophysical Research: Atmospheres*, 119(6):3372–3400, doi:10.1002/2013JD021119.

Korolev, A., Fugal, J., Krämer, M., McFarquhar, G., Lawson, P., Wendisch, M., Borrmann, S., Abel, S. J., Schnaiter, M., Franklin, C., Crosier, J., Williams, E., Wang, Z., Axisa, D., Lohmann, U., Field, P. R., and Schlenczek, O. (2017). Mixed-Phase Clouds: Progress and Challenges. *Meteorological Monographs*, 58(Fahrenheit 1724):5.1–5.50, doi:10.1175/amsmonographs-d-17-0001.1.

Lohmann, U. and Neubauer, D. (2018). The importance of mixed-phase and ice clouds for climate sensitivity in the global aerosol–climate model echam6-ham2. *Atmospheric Chemistry and Physics*, 18(12):8807–8828, doi:10.5194/acp-18-8807-2018.

Stevens, B., Giorgetta, M., Esch, M., Mauritsen, T., Crueger, T., Rast, S., Salzmann, M., Schmidt, H., Bader, J., Block, K., Brokopf, R., Fast, I., Kinne, S., Kornblueh, L., Lohmann, U., Pincus, R., Reichler, T., and Roeckner, E. (2013). Atmospheric component of the MPI-M earth system model: ECHAM6. *Journal of Advances in Modeling Earth Systems*, 5(2):146–172, doi:10.1002/jame.20015.

Zhang, K., O'Donnell, D., Kazil, J., Stier, P., Kinne, S., Lohmann, U., Ferrachat, S., Croft, B., Quaas, J., Wan, H., Rast, S., and Feichter, J. (2012). The global aerosol-climate model echam-ham, version 2: sensitivity to improvements in process representations. *Atmospheric Chemistry and Physics*, 12(19):8911–8949, doi:10.5194/acp-12-8911-2012.

[Figure]

Figure 1: Vertical profiles of cloud cover for winter and summer in the Arctic as well as the thresholds for the low/mid/high classification.

---

## Author Response (AR2)

**Response to Reviewer #2 and to the Co-Editor**

Firstly, we want to thank the two reviewers and the co-editor for thoroughly reading through the revised manuscript and again for providing comments on it. In the following, we address the comments of reviewer #2 and the co-editor. Additionally, all minor comments brought forward by the reviewers and the co-editor have been worked into the manuscript.

**Comparison to ground based observations**

**Reviewer #2:**

The authors are missing the point of my first comment I am aware that the authors used the COSP simulator, but my point is that since the focus of this study is on low-level clouds, the lowest 1-2 km of Arctic clouds will not be visible by GOCCP or the model with the COSP simulator on, whereas these clouds may exist according to ground-based observations (see the references listed in my original comment). Just because the COSP or GOCCP cannot "see" those low-clouds, does not mean that they are not actually there. Again, nothing prevents the authors from analyzing collocated Arctic ground-based observations to get a sense of the extent to which the claimed underestimate of low-cloud fraction actually holds, which was part of my original recommendation. At this point, Im still not convinced that this high bias in cloud cover exists in the first place.

**Co-Editor:**

Firstly, I agree that it is important to demonstrate that there is indeed a model bias in low-cloud cover and liquid cloud cover. I appreciate your new figure, but unfortunately, I don't quite see your logic. [...] It looks like you have done some analyses using ground-based measurements. Could you please explain what prevents you from doing a more direct comparison here?

Regarding the comparison of ground-based observations to modeled cloud fraction, we were reluctant to include that into the revised manuscript. As pointed out by reviewer #2 regarding our comparison of cloud phase fraction, any comparison between modeled and observed quantities easily is misleading if both are on different spatial scales. Additionally, it is intricate to make such a comparison fully consistent due to fundamental differences in the way physical properties are diagnosed in the model and in observations. This is due to for example finite observational detection thresholds, differences in sampling or even the physical representation of the relevant processes (Kay et al., 2016). It is precisely for these two reasons that in our study we compare the general circulation model to satellite data, namely that the spatial scales match, and that we have a satellite simulator implemented that allows for an apples-to-apples comparison.

We nevertheless performed a comparison to ground-based derived cloud fraction profiles for two sites in the Arctic, namely Barrow and Eureka, and compared the differences to the ECHAM6 native cloud cover as diagnosed directly by its cloud cover scheme. We consider this difference as the 'true' difference between modeled and observed clouds and we compared this to ECHAM6 + COSP/CALIPSO-GOCCP differences on which we based our claim of an overestimated low-level cloud fraction in the Arctic. The ground-based profiles are based on 35-GHz millimeter cloud radars (MMCR) in Barrow and Eureka for the years 2007 to 2009 as citet in Shupe et al. (2011). For better comparison, all datasets are vertically interpolated onto the ECHAM6 grid as it has the largest grid spacing in the vertical. To illustrate the spatial variability, we plotted profiles of cloud fraction differences in a  $3 \times 3$ and  $5 \times 5$  grid around the respective station (see Figure 1 and Figure 2). One might now argue that a  $5 \times 5$  grid or even a  $3 \times 3$  grid might not be representative if one compares it to spatially fixed observations, especially if environmental conditions vary as much as it is the case for Barrow and Eureka due to close vincinity of the sea with which we would definitely agree. In each numerical model, and in particular also in the spectral model ECHAM6, individual grid-points do not carry independent information, and the effective resolution is substantially coarser than the nominal resolution. ECHAM6 solves the primitive equations in spherical harmonics and artifacts of this are visible in the model output as 'wavy' structures (see plots of global cloud cover in Stevens et al. (2013) for examples of those 'wavy' structures). It is therefore necessary to choose a large enough observational area that is

as least as large as one wavelength of those 'wavy' structures to avoid any local minimum or maximum and to get statistically significant results.

In the following comparison, the ground-based profile is always the same while for gridded data, the grid point where the difference is evaluated is shifted within the  $n \times n$  grid. The solid line represents the median cloud fraction difference in the  $n \times n$  grid while the shaded area shows the range between the maximum and the minimum difference out of the  $n \times n$  grid for each vertical level. For the  $3 \times 3$  grid in Figure 1, the ECHAM6+COSP / CALIPSO-GOCCP difference shows a positive bias close to the surface except for Barrow in spring but here, the spatial variation is quite large as can be seen from the shaded area. One also sees that the difference between ECHAM6(native) and ground-based observation of cloud fraction is also positive near the surface except for Barrow in winter where it is slightly negative. We repeated the comparison on a  $5 \times 5$  grid (see Figure 2) and here, the median of both the ECHAM6 / ground-based and ECHAM6+COSP / CALIPSO-GOCCP difference is positive for Barrow and Eureka in winter and spring. As one might expect, the spatial variability increases for the  $5 \times 5$  grid as can be seen from the shaded range. For all stations and seasons, the median is rather on the high side than on the low side for cloud cover differences which makes us confident that cloud fraction is overestimated in the  $5 \times 5$  region around Barrow and Eureka.

The comparison shows that compared to ground-based observations, the model slightly overestimates cloud fraction in layers close to the surface, even though not as pronounced as it is the case for the ECHAM6+COSP / CALIPSO-GOCCP difference, especially for the  $3\times3$  grid. Additionally, as can be seen from the large variability of the ECHAM6+COSP / CALIPSO-GOCCP difference around those two station (where the evaluated grid point the shifted consistently within the two datasets), it becomes obvious that observations from Barrow and Eureka might not be representative for the whole Arctic, which is especially the case close to the surface both in the  $3\times3$  and the  $5\times5$  grid.

**DARDAR**

**Co-editor #2:**

Also, are you aware of the work presented in Forbes and Ahlgrimm (MWR, 2014)? They used DAR-DAR, which can help alleviate the limitation in Calipso.

The co-editor proposed to use DARDAR that combines the CloudSat radar and the CALIPSO lidar measurments instead of just using CALIPSO. DARDAR indeed could help to circumvent some of the observational idiosyncrasies of the lidar. As it was shown by Liu et al. (2017), making use of a combined radar-lidar product would especially be beneficial for the retrieval of clouds down to 1 km. Below that altitude, a combined radar-lidar product solely relies on the lidar data as the radar is affected by ground clutter close to the surface. Therefore, even DARDAR will underestimate the clouds close to the surface (as also the CALIPSO derived cloud fraction does) and probably the added benefit for clouds close to the surface would only be minor using DARDAR. What additionally prevented us from using a combined radar-lidar product was also the fact that there is presently no combined radarlidar simulator available within ECHAM6. As we have shown in Figure 1 in the revised manuscript, it is crucial to perform a definition aware comparison between observed and modeled clouds, especially close to the surface.

**Change of nadir angle in late 2007**

**Reviewer #2:**

I would recommend that the authors exclude the year 2007 and include another year (2011) to avoid the issue of the change in nadir viewing angle. That way, the comparison between model and observations would at least be more consistent.

**Co-editor #2:**

Secondly, regarding the data time periods used in your analysis, I understand the reviewer's concern.

This may have been taken care of in your GOCCP dataset, but it is better to double check with the data provider.

As brought up by reviewer #2, the nadir-pointing angle of CALIOP has changed in late 2007, which introduces an inconsistency into CALIPO cloud retrieval. A possible solution to circumvent that would be to not include the year 2007 in our analysis. The consequences of this measure would be that we would need to rerun all the simulation used in the manuscript and especially the sensitivity studies as only 2007 and 2008 are used. This would be a considerable amount of extra work and computing time which also can only be done later this year as we are presently over our quota at our computing center. Therefore, this would also further delay the completion of the revised manuscript.

To show the effects of the change in nadir angle on our evaluation of Arctic clouds in ECHAM6, we compared the ECHAM-COSP minus CALIPSO-GOCCP difference for 2 periods in the basic configuration of ECHAM6, one from 2007 to 2011 (Figure 3) and one from 2008-2012 (Figure 4). The comparison of the two plots shows that the differences in cloud cover and also in cloud phase are rather small and can be probably be attributed to internal variability and the results of our initial evaluation remain more or less unchanged.

As stated by Cesana et al. (2016), the change in the nadir-pointing angle resulted in less false cloud detection and less false liquid cloud determination since ice crystal plates produce the same signature as liquid droplets. One would therefore expect that difference in cloud fraction would further increase (less clouds in the GOCCP dataset) and that the bias in liquid clouds would also further increase (less liquid containing clouds in the GOCCP dataset). Even though one can identify those changes in Figure 3 and Figure 4, internal variability probably dominates over the effect of changing the viewing geometry which prevents us from drawing any conclusion in this regard.

**Relative humidity bias as a proxy for cloud cover bias**

**Reviewer #2:**

I see the logic of the authors in comparing profiles of temperature and relative humidity to the model to support their claim of the low-cloud overestimate, but because there is no clean causal relationship that can be teased apart, this additional piece of information is too speculative and not very useful in my opinion.

**Co-editor #2:**

The newly-added arguments (including the comparison in temperature and humidity), as mentioned in the reviewer's 4th bullet point, don't provide watertight evidence to support the existence of cloud cover biases.

Regarding the comment of reviewer #2 and the co-editor of no clean causal relationship between relative humidity and cloud cover, this at least is not the case for the way cloud cover is diagnosed in ECHAM6 as it uses relative humidity alone to diagnose fractional cloud cover. At least from the model side, high values of relative humidity are indicative of a high cloud cover. We understand that such a direct relationship between cloud cover and relative humidity might be oversimplified compared to how clouds are formed in the real atmosphere. Of course this should not be interpreted as a watertight evidence (which we never claimed it to be) but it supports our initial claim of overestimated cloud cover in high latitudes in ECHAM6 as we have shown using satellite observations. As we have also stated in the revised manuscript, one has to be careful when using ERA-Interim in higher latitudes due to the sparse availability of observations in high latitudes that constrain the reanalysis. In those regions, ERA-Interim is just another model so any conclusions drawn from such a comparison have to be taken with a grain of salt. This is the reason why we added the actual profiles observed by radiosondes that show the relative humidity in ERA-Interim is at least in a similar ballpark.

**Ice fraction**

**Reviewer #2:**

Comparison between Figure 6 and the figure from Korolev et al. (2017) is not meaningful because the latter observations were spatially averaged over 100 m, while the GCM grid box used in the manuscript has a horizontal resolution that is orders of magnitude larger. In fact, the authors seem to acknowledge the difference in spatial scales. A comparison with satellite observations would therefore be more of an apples to apples comparison. Furthermore, the plot over Siberia only adds to my initial concern that the WBF tuning exacerbates the phase partitioning problem by increasing ice fraction to values close to 1.0 between temperatures -5C and -35C for the lower threshold value plotted.

In her/his previous review, reviewer #2 asked to look into how cloud thermodynamic phase is affected by tuning of the WBF process. We assumed that the reviewer was demanding a mass-based phase fraction instead of a frequency ratio fraction of the top-cloud phase (that can be provided by CALIOP). A mass-based phase fraction cannot be provided by satellite observations which was the reason why we referred to Korolev et al. (2017), being aware that this might less-than-ideal dataset to compare it to a climate model due to different spatial scales. As reviewer #2 proposed, a satellite derived estimate of cloud phase would be better suited to compare it to a climate model, but here, we again end up in the dilemma that a mass-based phase fraction can not be provided by a satellite. A comparison of the frequency ratio fraction of the top-cloud phase fraction which can be provided by CALIOP is in principle nothing else than the comparison of liquid/ice cloud fraction we already have in our manuscript, so we would not get a different result from such a comparison.

Regarding the suspected phase partitioning problem mentioned by reviewer #2 in her/his present review, we clearly worked out in our last reply that ECHAM6, unlike many other climate models, actually has too much liquid water at low temperatures in the Arctic. We elaborated on that in the previous reply to reviewer #2 and we were able to show that the suspected phase partitioning issue brought forward by reviewer #2 does not hold for temperatures below  $-25^{\circ}$  C in high latitudes.

As we stated in the revised manuscript, we think that setting  $\gamma_{\text{thr}}$  to  $1 \cdot 10^{-6} \text{ kg m}^{-3}$  might already be a too extreme value and that it is in general hard (if not impossible) to tune cloud phase on a global scale with the present implementation of the cloud microphysics in ECHAM6.

**Scientific importance and reasons for the necessity of this study**

**Co-editor #2:**

Thirdly, while I appreciate a lot of work has been done in this manuscript and I believe your work will be valuable for the community, I do worry that these model developments have been applied to other models. It is unclear to me what is new in terms of our scientific understanding. To fit the scope of ACP, the new part needs to be shown more clearly to readers.

First and foremost, this study is by no means intended to be a tuning exercise of ECHAM6 but exploring the sensitivity to the representation of physical processes in ECHAM6. A main motivation for this study was to work out why ECHAM6/MPI-ESM differs in the representation of Arctic clouds compared to other CMIP5 models. We admit that we poorly motivated that in the introduction and a more elaborate motivation was added in the revised manuscript as well as clearer discussion of the results. To our knowledge, such a thorough analysis has not yet been performed for a GCM behaving like ECHAM6 and this study gives insight into processes that might be responsible for the behavior of the model.

**References**

- Cesana, G., Chepfer, H., Winker, D., Getzewich, B., Cai, X., Jourdan, O., Mioche, G., Okamoto, H., Hagihara, Y., Noel, V., and Reverdy, M. (2016). Using in situ airborne measurements to evaluate three cloud phase products derived from calipso. *Journal of Geophysical Research: Atmospheres*, 121(10):5788–5808, doi:10.1002/2015JD024334.
- Kay, J. E., L'Ecuyer, T., Chepfer, H., Loeb, N., Morrison, A., and Cesana, G. (2016). Recent advances in arctic cloud and climate research. *Current Climate Change Reports*, 2(4):159–169, doi:10.1007/s40641-016-0051-9.
- Korolev, A., Fugal, J., Krämer, M., McFarquhar, G., Lawson, P., Wendisch, M., Borrmann, S., Abel, S. J., Schnaiter, M., Franklin, C., Crosier, J., Williams, E., Wang, Z., Axisa, D., Lohmann, U., Field, P. R., and Schlenczek, O. (2017). Mixed-Phase Clouds: Progress and Challenges. *Meteorological Monographs*, 58(Fahrenheit 1724):5.1–5.50, doi:10.1175/amsmonographs-d-17-0001.1.
- Liu, Y., Shupe, M. D., Wang, Z., and MacE, G. (2017). Cloud vertical distribution from combined surface and space radar-lidar observations at two Arctic atmospheric observatories. *Atmospheric Chemistry and Physics*, 17(9):5973–5989, doi:10.5194/acp-17-5973-2017.
- Shupe, M. D., Walden, V. P., Eloranta, E., Uttal, T., Campbell, J. R., Starkweather, S. M., and Shiobara, M. (2011). Clouds at Arctic atmospheric observatories. Part I: Occurrence and macrophysical properties. *Journal of Applied Meteorology and Climatology*, 50(3):626–644, doi:10.1175/2010JAMC2467.1.
- Stevens, B., Giorgetta, M., Esch, M., Mauritsen, T., Crueger, T., Rast, S., Salzmann, M., Schmidt, H., Bader, J., Block, K., Brokopf, R., Fast, I., Kinne, S., Kornblueh, L., Lohmann, U., Pincus, R., Reichler, T., and Roeckner, E. (2013). Atmospheric component of the MPI-M earth system model: ECHAM6. Journal of Advances in Modeling Earth Systems, 5(2):146–172, doi:10.1002/jame.20015.

Figure 1: Cloud fraction profile differences evaluated for gridpoints on a  $3 \times 3$  grid around the respective station. The solid line represents the median of the differences on the  $3 \times 3$  grid.